# ST-DDPM: Explore Class Clustering for Conditional Diffusion Probabilistic Models

## Abstract

Score-based generative models involve sequentially corrupting the data distribution with noise and then learns to recover the data distribution based on score matching. In this paper, for the diffusion probabilistic models, we first delve into the changes of data distribution during the forward process of the Markov chain and explore the class clustering phenomenon. Inspired by the class clustering phenomenon, we devise a novel conditional diffusion probabilistic model by explicitly modeling the class center in the forward and reverse process, and make an elegant modification to the original formulation, which enables controllable generation and gets interpretability. We also provide another direction for faster sampling and more analysis of our method. To verify the effectiveness of the formulated framework, we conduct extensive experiments on multiple tasks, and achieve competitive results compared with the state-of-the-art methods (conditional image generation on CIFAR-10 with an inception score of 9.58 and FID score of 3.05).

## 1 Introduction

Deep generative models such as generative adversarial networks (GANs) Goodfellow et al. (2014); Zhu et al. (2017); Brock et al. (2018), flows Rezende & Mohamed (2015); Kingma & Dhariwal (2018); Ho et al. (2019), variational autoencoders (VAEs) Kingma & Welling (2013); Maaløe et al. (2019), score-based generative models Song & Ermon (2019); Ho et al. (2020); Song et al. (2020b) and autoregressive models Van Oord et al. (2016); Oord et al. (2016); Salimans et al. (2017) have been proposed to generate high quality samples in wide variety of data modalities. The score-based generative models involve sequentially corruptig the data distribution with noise and then learns to recover the data distribution based on score matching (e.g. the gradient of the log probility density Song & Ermon (2019); Song et al. (2020b), the noise corruption Ho et al. (2020)).

Given the Markov chain that gradually corrupts the data with noise, in this paper, we first delve into the changes of data distribution during the forward process and visualize these behaviours. Specifically, we measure the class clustering by considering the intra-class to inter-class variance ratio at each time step. Low values of the variance ratio demonstrate better class separation since the samples are concentrated around their corresponding class mean. We notice that in the early stage of forward process, the data distribution maintains stable intra-class variance and inter-class variance, leading to a balanced variance ratio. As the noise gradually increases, the class separation decreases sharply and the data distribution is ultimately converted into the noise distribution, as shown in Figure 1 (Left). The class clustering phenomenon reveals the process of diffusion where the corruption noise first corrupts the datapoints without increasing the variance ratio, and then greatly decreases the class separation and mixes the datapoints from different classes, which generates the noise distribution.

Inspired by the class clustering phenomenon, we formulate a novel conditional diffusion probabilistic model by explicitly modeling the class center in the forward and reverse process, which enables controllable generation. Concretely, as shown in Figure 1 (Right), in the early stage of forward process, samples are gradually clustered to their corresponding class center while all the samples are finally gathered into the noise distribution as the diffusion progresses. The reverse process is then modulated by the class center during generation:

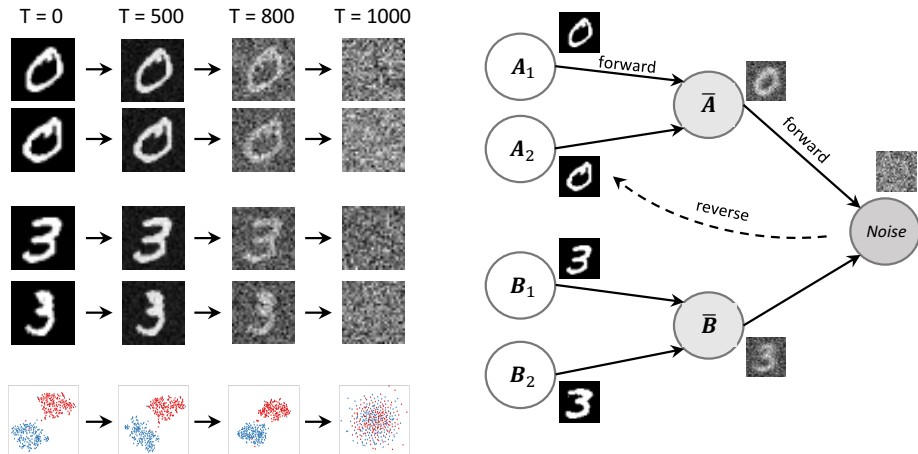

Figure 1: **Left:** An example of the class clustering phenomenon. **Right:** The overview of the proposed conditional diffusion probabilistic model where $A_i$, $B_i$ are samples from different classes, and $\bar{A}$, $\bar{B}$ are the class centers.

1. By explicitly introducing the class center, we make an elegant modification to the original formulation Ho et al. (2020) and further decouple the module for conditional information encoding and the module for noise prediction. The formulation that exploits the class center is much simpler since we don't need to train a separate model or apply heuristics and domain knowledge to estimate the gradient of log likelihood Song et al. (2020b).

2. The learned class centers, which get interpretability, also reflect the common characteristics of datapoints from the same class (e.g. template face, number).

3. With the explicitly-modeled class centers, we can naturally speed up the reverse process with earlier starting, which is different from strided sampling Song et al. (2020a).

Due to the guided "shift" from sample to its corresponding class center during both the forward and reverse process, we name the conditional DDPM as ST-DDPM. To verify the effectiveness of the formulated framework ST-DDPM, we conduct extensive experiments on the task of conditional image generation, free-form image inpainting Yu et al. (2019); Liu et al. (2018), attribute-to-image synthesis Yan et al. (2016) and text-to-image synthesis Reed et al. (2016); Xu et al. (2018); Zhang et al. (2018), and achieve competitive results compared with the original formulation. We also provide another direction for faster sampling and further explore the relationship between the guided class centers and other score-based methods.

## 2 RELATED WORK

**Score-based Generative Models.** Score-based generative models Song & Ermon (2019); Ho et al. (2020); Song et al. (2020b); Tae et al. (2021); Kim et al. (2021); Tashiro et al. (2021) involve sequentially corruptig the data distribution with noise using Markov chains and then learns to convert the noise distribution to the data distribution based on score matching. Score matching with Langevin dynamics (SMLD) Song & Ermon (2019) learns to estimate the gradient of the log probility density with respect to data at each time step and samples from a sequence of decreasing noise scales for generation. Denoising diffusion probabilistic model (DDPM) Ho et al. (2020) trains the model with denoising score matching to predict the noise corruption at each step and then reverse the noise distribution during generation. Song et al. Song et al. (2020b) further propose a unified framework to generalize prior works by introducing the stochastic differential equations (SDEs). There are other methods to learn the reverse process of Markov chains including infusion training Bordes et al. (2017), variational walkback Goyal et al. (2017), generative stochastic networks Alain et al. (2016) and others Salimans et al. (2015); Song et al. (2017); Levy et al. (2017); Nijkamp et al. (2019); Lawson et al. (2019).

## 3 DENOISING DIFFUSION PROBABILISTIC MODELS

The DDPM Ho et al. (2020) employs a forward diffusion Markov chain to step-by-step convert a real data distribution into a standard Gaussian distribution, and then build a parameterized Markov chain to model the reverse process. Specifically, the forward process starts from $\mathbf{x}_0$ which comes from the real data distribution $q(\mathbf{x}_0)$, and performs $T$ steps of diffusion to generate $\mathbf{x}_1, \mathbf{x}_2, \cdots, \mathbf{x}_T$ sequentially. The forward trajectory is given by

$$q(\mathbf{x}_{0:T}) = q(\mathbf{x}_0) \prod_{t=1}^{T} q(\mathbf{x}_t|\mathbf{x}_{t-1}) \qquad q(\mathbf{x}_t|\mathbf{x}_{t-1}) = \mathcal{N}(\mathbf{x}_t; \sqrt{1-\beta_t}\mathbf{x}_{t-1}, \beta_t\mathbf{I}) \qquad (1)$$

where $\beta_1, \beta_2, \cdots, \beta_T$ correspond to aforementioned diffusion rate of each forward step and are fixed as a variance schedule. The reverse process is conducted by parameterized operators $p_\theta(\mathbf{x}_{t-1}|\mathbf{x}_t)$:

$$p_\theta(\mathbf{x}_{0:T}) = p(\mathbf{x}_T) \prod_{t=1}^{T} p_\theta(\mathbf{x}_{t-1}|\mathbf{x}_t) \qquad p_\theta(\mathbf{x}_{t-1}|\mathbf{x}_t) = \mathcal{N}(\mathbf{x}_{t-1}; \boldsymbol{\mu}_\theta(\mathbf{x}_t, t), \boldsymbol{\Sigma}_\theta(\mathbf{x}_t, t)) \qquad (2)$$

where $p(\mathbf{x}_T) = \mathcal{N}(\mathbf{x}_T; \mathbf{0}, \mathbf{I})$. The DDPM is trained to minimize a variational bound of negative log likelihood $\mathbb{E}_q[-\log p_\theta(\mathbf{x}_0)]$, denoted by:

$$\mathbb{E}_q[-\log p_\theta(\mathbf{x}_0)] \le \mathbb{E}_q\left[-\log p(\mathbf{x}_T) - \sum_{t=1}^{T} \log \frac{p_\theta(\mathbf{x}_{t-1}|\mathbf{x}_t)}{q(\mathbf{x}_t|\mathbf{x}_{t-1})}\right] := L.$$

Rewrite $L$ (3) by Bayes rule establishes a connection between $p_\theta(\mathbf{x}_{t-1}|\mathbf{x}_t)$ and $q(\mathbf{x}_{t-1}|\mathbf{x}_t, \mathbf{x}_0)$ for backward operators:

$$L = \mathbb{E}_q\left[-\log \frac{p(\mathbf{x}_T)}{q(\mathbf{x}_T|\mathbf{x}_0)} - \sum_{t=2}^{T} \log \frac{p_\theta(\mathbf{x}_{t-1}|\mathbf{x}_t)}{q(\mathbf{x}_{t-1}|\mathbf{x}_t, \mathbf{x}_0)} - \log p_\theta(\mathbf{x}_0|\mathbf{x}_1)\right] \qquad (3)$$

$$= \mathbb{E}_q\left[KL(q(\mathbf{x}_T|\mathbf{x}_0) \parallel p(\mathbf{x}_T)) + \sum_{t=2}^{T} KL(q(\mathbf{x}_{t-1}|\mathbf{x}_t, \mathbf{x}_0) \parallel p_\theta(\mathbf{x}_{t-1}|\mathbf{x}_t)) - \log p_\theta(\mathbf{x}_0|\mathbf{x}_1)\right], \qquad (4)$$

where the forward process posterior conditioned upon $\mathbf{x}_0$ is related to the forward process distribution via Bayes rule, denoted by:

$$q(\mathbf{x}_{t-1}|\mathbf{x}_t, \mathbf{x}_0) = \frac{q(\mathbf{x}_t|\mathbf{x}_{t-1}, \mathbf{x}_0)q(\mathbf{x}_{t-1}|\mathbf{x}_0)}{q(\mathbf{x}_t|\mathbf{x}_0)}.$$

Fortunately, all right items have closed form expressions, which can exactly derive a Gaussian distribution, denoted by:

$$q(\mathbf{x}_{t-1}|\mathbf{x}_t, \mathbf{x}_0) = \mathcal{N}(\mathbf{x}_{t-1}; \frac{\sqrt{\bar{\alpha}_{t-1}}\beta_t}{1-\bar{\alpha}_t}\mathbf{x}_0 + \frac{\sqrt{\alpha_t}(1-\bar{\alpha}_{t-1})}{1-\bar{\alpha}_t}\mathbf{x}_t, \frac{1-\bar{\alpha}_{t-1}}{1-\bar{\alpha}_t}\beta_t\mathbf{I}),$$

where $\alpha_t = 1 - \beta_t$ and $\bar{\alpha}_t = \prod_{i=1}^{t} \alpha_i$. Consequently, the KL divergences between Gaussians in $L$ (4) can be computed analytically. Further simplications and reparameterizations Ho et al. (2020) come to the following variant of the variational bound:

$$L_{simple}(\theta) = \mathbb{E}_{t,\mathbf{x}_0,\epsilon}\left[\parallel \epsilon - \epsilon_\theta(\sqrt{\bar{\alpha}_t}\mathbf{x}_0 + \sqrt{1-\bar{\alpha}_t}\epsilon, t) \parallel^2\right]$$

## 4 MEASURING CLASS CLUSTERING IN RGB AND FEATURE SPACES

We first delve into the changes of data distribution during the forward process and visualize these behaviours. Concretely, we measure the class clustering in RGB and feature spaces by considering the intra-class to inter-class variance ratio at each time step, given by

$$\frac{\sigma_{intra-class}^2}{\sigma_{inter-class}^2} = \frac{C}{N} \frac{\sum_{i,j} \|x_{i,j} - \mu_i\|_2}{\sum_i \|\mu - \mu_i\|_2} \qquad (5)$$

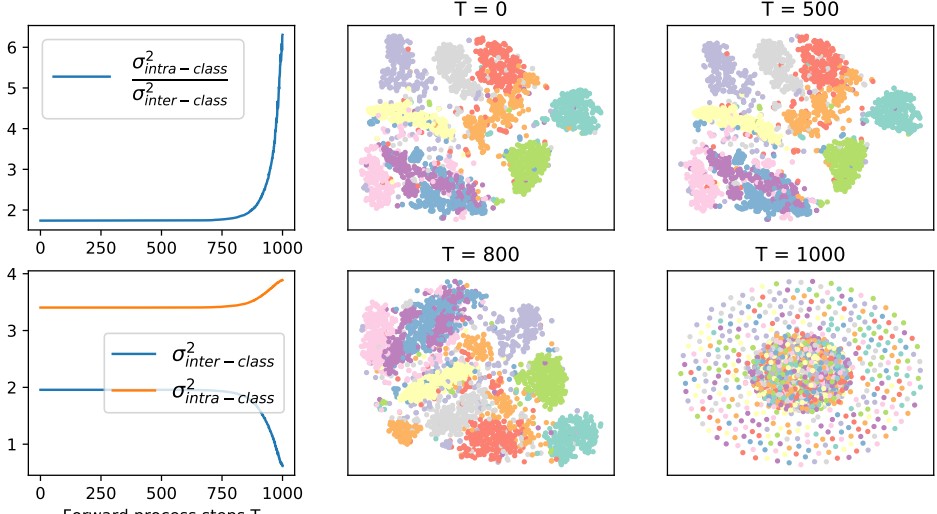

Figure 2: The changes of data distribution and class clustering in RGB space on MNIST dataset.

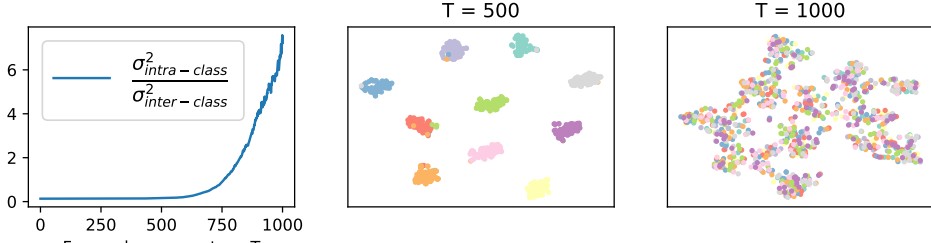

Figure 3: The changes of data distribution and class clustering in feature space on CIFAR-10 dataset.

where $C$ is the number of classes, $N$ is the number of data points, $x_{i,j}$ is the $j$-th sample in class $i$, $\mu_i$ is the mean of samples of class $i$ and $\mu$ is the mean across all the samples. Low values of the variance ratio demonstrate better class separation. We compute the variance ratio and visualize the data distribution on MNIST and CIFAR-10 dataset. Note that for better visualization, the feature vectors for CIFAR-10 dataset are computed based on pre-trained ResNet18 backbone.

As shown in Figure 2 and Figure 3 given by t-SNE visualization, in the early stage of forward process, the data distribution maintain stable intra-class variance and inter-class variance, leading to a balanced class clustering (T approximately from 0 to 800). As the noise gradually increases, the class separation decreases sharply and the data distribution is ultimately converted into the noise distribution (T = 1000). The class clustering phenomenon inspires us to explicitly introduce the class center or class template during forward and reverse process for controllable generation.

## 5 THE FORMULATION OF ST-DDPM

Inspired by the class clustering phenomenon, we formulate the conditional diffusion probabilistic model ST-DDPM by explicitly modeling the condition as the class center in the forward and reverse process. Concretely, we introduce the condition encoding network to obtain the class center or class template as $\mathbf{u} = \mathbf{u}_\phi(\mathbf{c})$, where $\mathbf{c}$ is the given condition (e.g. corrupted image, class label or attributes), and then rebuild the diffusion operators and modify the original recurrence relation Eq. (1), given by:

$$q(\mathbf{x}_t|\mathbf{x}_{t-1}, \mathbf{u}) = \mathcal{N}(\mathbf{x}_t; \sqrt{\alpha_t}\mathbf{x}_{t-1} + \bar{\alpha}_t^{\frac{1}{4}}(1-\alpha_t^{\frac{1}{4}})\mathbf{u}, \beta_t\mathbf{I}) \qquad \mathbf{x}_t = \sqrt{\alpha_t}\mathbf{x}_{t-1} + \bar{\alpha}_t^{\frac{1}{4}}(1-\alpha_t^{\frac{1}{4}})\mathbf{u} + \sqrt{\beta_t}\boldsymbol{\epsilon}_t,$$

$$(6)$$

where we add a condition-related item for Gaussian mean to make the forward process has a "shift" toward the class center. With the recurrence relation, we can derive the general $\mathbf{x}_t$ sampling formula:

$$\mathbf{x}_t = \sqrt{\bar{\alpha}_t}\mathbf{x}_0 + \bar{\alpha}_t^{\frac{1}{4}}(1 - \bar{\alpha}_t^{\frac{1}{4}})\mathbf{u} + \sqrt{1 - \bar{\alpha}_t}\boldsymbol{\epsilon}_t = \bar{\alpha}_t^{\frac{1}{4}}\left[\bar{\alpha}_t^{\frac{1}{4}}\mathbf{x}_0 + (1 - \bar{\alpha}_t^{\frac{1}{4}})\mathbf{u}\right] + \sqrt{1 - \bar{\alpha}_t}\boldsymbol{\epsilon}_t. \tag{7}$$

---

**Algorithm 1:** Training

---

**repeat**

    $\mathbf{x}_0 \sim q(\mathbf{x}_0)$

    $\mathbf{u} = \mathbf{u}_\phi(\mathbf{c})$

    $t \sim Uniform(1, 2, \cdots, T)$

    $\boldsymbol{\epsilon} \sim \mathcal{N}(\mathbf{0}, \mathbf{I})$

    Optimize $\| \boldsymbol{\epsilon} - \left[\boldsymbol{\epsilon}_\theta(\sqrt{\bar{\alpha}_t}\mathbf{x}_0 + \mathbf{n}_t + \sqrt{1 - \bar{\alpha}_t}\boldsymbol{\epsilon}, t) - \frac{\mathbf{n}_t}{\sqrt{1 - \bar{\alpha}_t}}\right] \|^2$

**until** *converged*;

---

**Algorithm 2:** Sampling

---

$\mathbf{x}_T \sim \mathcal{N}(\mathbf{0}, \mathbf{I})$

**for** $t = T$ **to** $1$ **do**

    $\boldsymbol{\epsilon} \sim \mathcal{N}(\mathbf{0}, \mathbf{I})$ if $t \geq 2$, else $\boldsymbol{\epsilon} = \mathbf{0}$

    $\mathbf{x}_{t-1} = \frac{1}{\sqrt{\alpha_t}}\left[\mathbf{x}_t - \frac{\beta_t}{\sqrt{1 - \bar{\alpha}_t}}\left[\boldsymbol{\epsilon}_\theta(\mathbf{x}_t, t) - \frac{\mathbf{n}_t}{\sqrt{1 - \bar{\alpha}_t}}\right] - \mathbf{m}_t\right] + \sqrt{\frac{\beta_t(1 - \bar{\alpha}_{t-1})}{(1 - \bar{\alpha}_t)}}\boldsymbol{\epsilon}$

**return** $\mathbf{x}_0$

---

See Appendix A.1 for detailed derivation. An intuitive explanation of the proposed diffusion operators and the weight-scheduling of $\mathbf{u}$ can be given as follows: Since $\bar{\alpha}_t$ is closer to 1 in the early stage, the class-guided forward process first corrupts the sample while maintains the specificity of samples. As the diffusion procedures, the samples from the same class gather at the class center. As $\bar{\alpha}_t$ gets smaller and closer to 0, the data distribution is ultimately converted into noise distribution. In total, inspired by the class clustering, the well-designed weights firstly allow the specificity of datapoints, then gather datapoints from the same class for universality, and finally turn the distribution into pure noise by eliminating the impact of class means.

For convenience, we denote $\bar{\alpha}_t^{\frac{1}{4}}(1 - \alpha_t^{\frac{1}{4}})\mathbf{u} = \mathbf{m}_t$ and $\bar{\alpha}_t^{\frac{1}{4}}(1 - \bar{\alpha}_t^{\frac{1}{4}})\mathbf{u} = \mathbf{n}_t$. Then the forward process posterior distribution $q(\mathbf{x}_{t-1}|\mathbf{x}_t, \mathbf{x}_0, \mathbf{u})$ can be derived from probability density:

$$q(\mathbf{x}_{t-1}|\mathbf{x}_t, \mathbf{x}_0, \mathbf{u}) = \mathcal{N}(\mathbf{x}_{t-1}; \boldsymbol{\mu}_t(\mathbf{x}_t, \mathbf{x}_0, \mathbf{u}), \sigma_t^2\mathbf{I}), \tag{8}$$

where

$$\boldsymbol{\mu}_t(\mathbf{x}_t, \mathbf{x}_0, \mathbf{u}) = \frac{\sqrt{\alpha_t}(1 - \bar{\alpha}_{t-1})}{1 - \bar{\alpha}_t}\mathbf{x}_t + \frac{\beta_t\sqrt{\bar{\alpha}_{t-1}}}{1 - \bar{\alpha}_t}\mathbf{x}_0 - \sqrt{\alpha_t}\frac{1 - \bar{\alpha}_{t-1}}{1 - \bar{\alpha}_t}\mathbf{n}_t + \mathbf{n}_{t-1} \quad \sigma_t^2 = \frac{\beta_t(1 - \bar{\alpha}_{t-1})}{(1 - \bar{\alpha}_t)}. \tag{9}$$

See Appendix A.2 for detailed derivation. Then we can directly model parameterized $\boldsymbol{\mu}_\theta(\mathbf{x_t}, \mathbf{u}, t)$ to predict $\boldsymbol{\mu}_t(\mathbf{x}_t, \mathbf{x}_0, \mathbf{u})$. Similar to Ho et al. (2020), we further reparameterize Eq. (7) for $\boldsymbol{\epsilon}_t \sim \mathcal{N}(\mathbf{0}, \mathbf{I})$, then the forward process posterior mean becomes:

$$\boldsymbol{\mu}_t\left(\mathbf{x}_t, \frac{\mathbf{x}_t - \mathbf{n}_t - \sqrt{1 - \bar{\alpha}_t}\boldsymbol{\epsilon}_\theta(\mathbf{x}_t, \mathbf{u}, t)}{\sqrt{\bar{\alpha}_t}}, \mathbf{u}\right) = \frac{1}{\sqrt{\alpha_t}}\left[\mathbf{x}_t - \frac{\beta_t}{\sqrt{1 - \bar{\alpha}_t}}\boldsymbol{\epsilon}_\theta(\mathbf{x}_t, \mathbf{u}, t) - \mathbf{m}_t\right]. \tag{10}$$

We then optimize the simplified loss to predict the corruption noise $\boldsymbol{\epsilon}$:

$$L_{simple}(\phi, \theta) = \mathbb{E}_{t, \mathbf{x}_0, \boldsymbol{\epsilon}}\left[\| \boldsymbol{\epsilon} - \boldsymbol{\epsilon}_\theta(\sqrt{\bar{\alpha}_t}\mathbf{x}_0 + \mathbf{n}_t + \sqrt{1 - \bar{\alpha}_t}\boldsymbol{\epsilon}, \mathbf{u}, t) \|^2\right] \tag{11}$$

Since we explicitly model the class center during the forward process, from Eq. (7), we have

$$\boldsymbol{\epsilon}_t = \frac{\mathbf{x}_t - \sqrt{\bar{\alpha}_t}\mathbf{x}_0}{\sqrt{1 - \bar{\alpha}_t}} - \frac{\mathbf{n}_t}{\sqrt{1 - \bar{\alpha}_t}}, \tag{12}$$

where we can extract $\mathbf{u}$ out of parameters and use $\boldsymbol{\epsilon}_\theta(\mathbf{x}_t, t)$ to predict $\frac{\mathbf{x}_t - \sqrt{\bar{\alpha}_t}\mathbf{x}_0}{\sqrt{1 - \bar{\alpha}_t}}$, and decouple the condition encoding module $\mathbf{u}_\phi$ and the denoising module $\boldsymbol{\epsilon}_\theta$, making the final simplified loss $L_{simple}(\phi, \theta)$ becomes:

$$L_{simple}(\phi, \theta) = \mathbb{E}_{t, \mathbf{x}_0, \boldsymbol{\epsilon}}\left\{\| \boldsymbol{\epsilon} - \left[\boldsymbol{\epsilon}_\theta(\sqrt{\bar{\alpha}_t}\mathbf{x}_0 + \mathbf{n}_t + \sqrt{1 - \bar{\alpha}_t}\boldsymbol{\epsilon}, t) - \frac{\mathbf{n}_t}{\sqrt{1 - \bar{\alpha}_t}}\right] \|^2\right\} \tag{13}$$

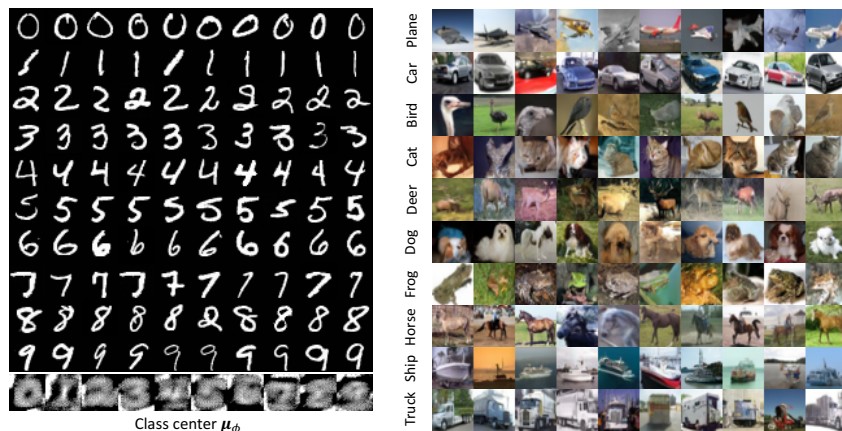

Figure 4: **Left:** Class-conditional samples on 28×28 MNIST and the trained conditional encoding network $\mathbf{u}_\phi$ (last row). **Right:** Class-conditional samples on 32×32 CIFAR-10.

Algorithm 1 and Algorithm 2 separately display the complete training procedure with this simplified objective and sampling procedure with the proposed ST-DDPM.

# 6 EXPERIMENTS

To verify the effectiveness of the formulated framework ST-DDPM, we conduct extensive experiments on the task of conditional image generation, free-form image inpainting Yu et al. (2019); Liu et al. (2018) and text-to-image synthesis Reed et al. (2016); Xu et al. (2018); Zhang et al. (2018). Following prior work Ho et al. (2020), we set $T = 1000$ and set the diffusion rate increasing linearly from $\beta_1 = 10^{-4}$ to $\beta_T = 0.02$ for all experiments. For the denoising network $\epsilon_\theta$, we use the same architecture as Ho et al. (2020), which is a U-Net based on a Wide ResNet since we focus on the impact of our formulation. More experimental details can be found in the Appendix B.

## 6.1 CLASS-CONDITIONAL IMAGE GENERATION

For class-conditional image generation, the condition refers to an one-hot vector corresponding to a specific class label. To consturct the condition encoding network $\mathbf{u}_\phi$, we map the one-hot vector to an initialized embedding and employ stacked convolution and upsample layers to predict the class center.

Figure 4 (Left) presents the class-conditional synthesis results and the learned class centers on the 28×28 MNIST dataset. As shown in the last row, the learned class centers also exhibit the shape of corresponding numbers, further indicating the class clustering phenomenon mentioned in the previous section. Figure 4 (Right) presents the conditional synthesis results on the

Table 1: CIFAR-10 sample quality.

| Model | FID↓ | IS↑ |
|---|---|---|
| **Unconditional** | | |
| NCSN Song & Ermon (2019) | 25.32 | 8.87 ± 0.12 |
| NCSNv2 Song & Ermon (2020) | 10.87 | 8.40 ± 0.07 |
| DDPM Ho et al. (2020) | 3.17 | 9.46 ± 0.11 |
| NCSN++ cont. (deep, VE) Song et al. (2020b) | 2.20 | 9.89 |
| **Conditional** | | |
| Projection Discriminator Miyato & Koyama (2018) | 17.5 | 8.62 |
| FQ-GAN Zhao et al. (2020) | 5.34 | 8.50 |
| BigGAN Brock et al. (2018) | 14.73 | 9.22 |
| grad. DDPM | 4.25 | 9.18 |
| cond. DDPM | 3.82 | 9.40 |
| ST-DDPM | **3.05** | **9.58 ± 0.09** |

32×32 CIFAR-10 Krizhevsky et al. (2009) dataset, also verifying the capability and effectiveness of our formulated method for controllable generation. The results of negative log likelihoods are presented in Appendix C and more examples can be found in the Appendix E.

To explore the sample quality, we compute the Inception scores and FID scores on CIFAR-10 dataset, as shown in Table 1. For comparison, following Song et al. (2020b), we introduce a time-dependent classifier and compute the gradient for class-conditional sampling (grad. DDPM). Also, we build a simple conditional DDPM (cond. DDPM) by directly injecting a class embedding along with the

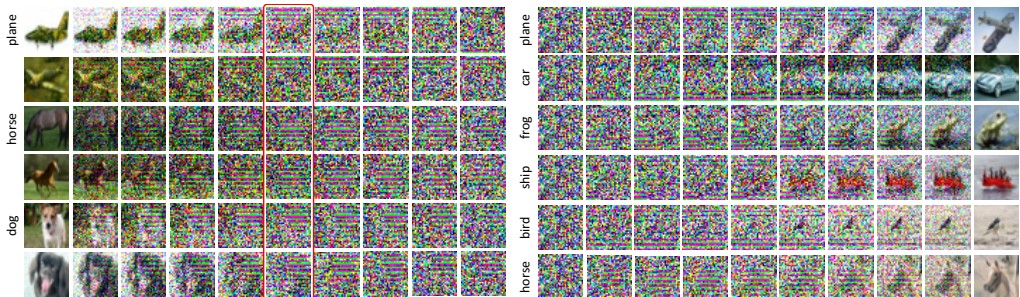

Figure 5: **Left:** The progressive diffusion modulated by the class center $\mathbf{u}_\phi$ on CIFAR-10 dataset. **Right:** The progressive decoding with the same intialized noise modulated by different class centers $\mathbf{u}_\phi$ on CIFAR-10 dataset.

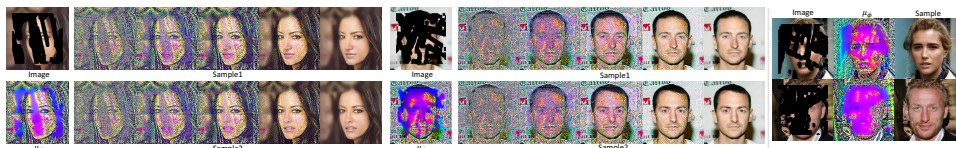

Figure 6: **Left:** The progressive decoding modulated by the class center $\mathbf{u}_\phi$ on 256×256 CelebA-HQ dataset. **Right:** More examples and corresponding class center $\mathbf{u}_\phi$.

timestep embedding. Details are given in the Appendix B. With the same denoising architecture as the unconditional DDPM Ho et al. (2020), the proposed ST-DDPM even obtains better IS score of 9.58 and FID score of 3.05. For class-conditional generation, our model achieves competitive results with the state-of-the-art methods, and better sample quality than most methods in the literature, indicating the effectiveness of the proposed method.

As shown in Figure 5, we further visualize the progressive diffusion and decoding of ST-DDPM modulated by different class centers. The ST-DDPM explicitly models the class center during the forward and reverse process, making the class clustering more significant. Figure 5 (Left) presents the progressive diffusion where the class clustering phenomenon is marked by the red bounding box. Figure 5 (Right) shows the progressive decoding toward different classes with the same initialized noise, qualitatively verifying the capability of ST-DDPM for controllable generation.

## 6.2 IMAGE INPAINTING

Image inpainting Yu et al. (2019); Liu et al. (2018) aims to synthesize contents for the missing regions of a given image such that the result is visually realistic and semantically correct. Different from class-conditional generation where the conditional encoding network is confined to a limited and discrete space (i.e. the number of class labels), the condition space of image inpainting is continuous and non-enumerable. To further prove the effectiveness of ST-DDPM that explicitly models the class center, we consider different images for imputation as different class and build a continuous conditional encoding network to generate the class center. Concretely, we denote the corrupted image as the input condition and apply a U-Net based architecture to predict its corresponding class center, making the class centers non-enumerable. The reverse process also starts from noise and is modulated by the predicted center.

Figure 7 separately presents the inpainting results on the 256×256 Place2 Zhou et al. (2017), 256×256 CelebA-HQ Liu et al. (2015) and 256×256 LSUN church Yu et al. (2015) dataset, which indicates the effectiveness and scalability of the ST-DDPM method. More examples can be found in the Appendix E. We also visualize the progressive diffusion decoding and the predicted class center in Figure 6. Notice that the decoupled condition encoding network has learned to predict the template face (i.e. $\mathbf{u}_\phi$) based on uncorrupted region, and the decoding process is modulated by the class center for inpainting, further demonstrating the effectiveness and interpretability of our proposed ST-DDPM. The further evaluation results can be found in Appendix C.

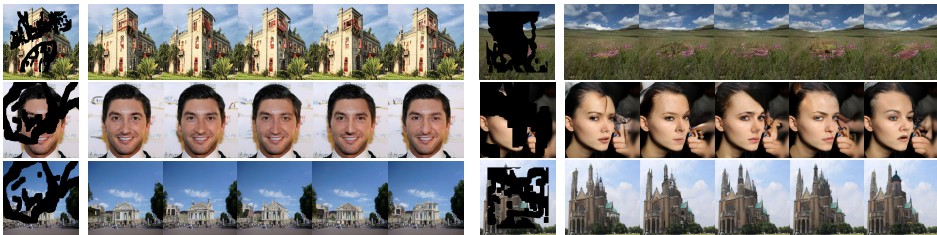

Figure 7: Inpainting results with different initialized noise on 256×256 Place2 dataset (first row), 256×256 CelebA-HQ dataset (second row) and 256×256 LSUN church dataset (third row).

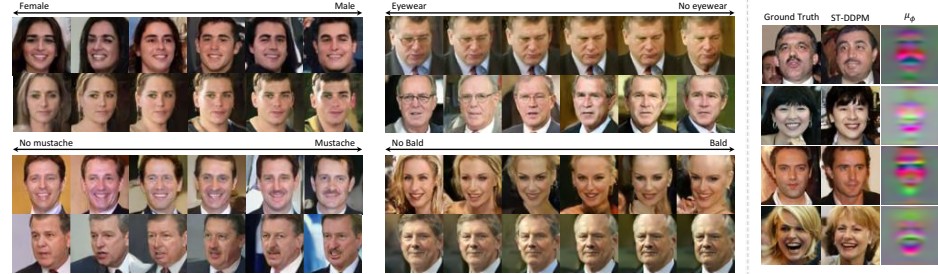

Figure 8: Attribute-to-image synthesis results on the 64×64 LFW dataset. (**Left**: Image progression conditioned on attribute (Interpolations on the attribute vectors). **Right**: Visualization of class centers $\mathbf{u}_\phi$ for test subset.)

## 6.3 ATTRIBUTE-TO-IMAGE SYNTHESIS

Attribute-to-image synthesis Yan et al. (2016) requires to generate object image from high-level visual attributes (e.g. age, gender, lighting), which are also continuous values. Similarly, we map the attribute vectors to class center (i.e. template face) by condition encoding network and leverage the proposed ST-DDPM formulation. By interpolating between the minimum and maximum attribute value to modify the value progressively, we can further explore the capabilities of ST-DDPM for image synthesis in a continuous condition space.

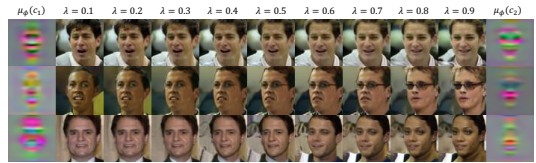

Figure 9: Interpolations on the condition space for the 64×64 LFW dataset.

Figure 8 presents the attribute-to-image synthesis results on the 64×64 LFW Huang et al. (2008) dataset. As shown in Figure 8 (Left), samples generated by progressive condition are visually consistent with attribute description, indicating that our method can effectively explore the continuous condition space. Figure 8 (Right) visualizes the class centers predicted by the condition encoding network and the $\mathbf{u}_\phi$ also shows the shape of template face, demonstrating the interpretability of our method. Since we decouple the condition encoding network and denoising network, we can further explore the condition space by interpretation of two different conditions, given by $\lambda \cdot \mathbf{u}_\phi(\mathbf{c}_1) + (1 - \lambda) \cdot \mathbf{u}_\phi(\mathbf{c}_2)$. Figure 9 shows the samples generated by interpolations on the condition space. The interpretation smoothly introduces attributes from two different conditions, which can be extended to multi-condition generation in the future.

## 6.4 TEXT-TO-IMAGE SYNTHESIS

Text-to-image synthesis Reed et al. (2016); Xu et al. (2018); Zhang et al. (2018) aims to generate realistic and text-consistent images according to the given natural language descriptions. We extend the ST-DDPM to the more challenging cross-modal generation by mapping the encoded word

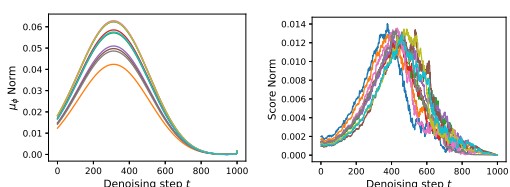

Figure 10: **Left:** Speedup on the 32×32 CIFAR-10 dataset for different time steps. **Right:** Faster diffusion denoising on the 32×32 CIFAR-10 dataset with different time steps.

sequences to class centers (details in the appendix B). Also, we obtain an Inception score of 4.52 and FID score of 18.18, demonstrating the effectiveness of our ST-DDPM method. More generated examples and visualization results of class centers can be found in the Appendix E.

### 6.5 FASTER DIFFUSION DENOISING

Different from strided sampling Song et al. (2020a), since we decouple the denoising network and condition encoding network and the diffusion process is modulated by the class center and datapoints from the same class gather earlier, as shown in Figure 5 (Left), we can devise a way for faster diffusion denoising that starts from earlier starting points $\mathbf{x}_T = \mathbf{n}_T + \boldsymbol{\epsilon}$, where $\boldsymbol{\epsilon} \sim \mathcal{N}(\mathbf{0}, \mathbf{I})$ (details in the appendix B). By reducing the number of reverse steps, we enable faster sampling without harming quality as much as possible.

Figure 10 presents the speedup and generated examples on the 32×32 CIFAR-10 dataset with different time steps. With $T = 700$, our method can achieve a speedup of $1.43\times$ and the sample quality is slightly decreased. With $T = 900$, our method can achieve a speedup of $1.12\times$ without harming the sample quality. Also, our technique of earlier starting (ES) can be combined with strided sampling (SS) and achieve a speedup of $11.1\times$, and detailed derivation and more results are given in the Appendix D.

### 6.6 SCORE-BASED METHODS FOR CONTROLLABLE GENERATION

With bayes rule, Song et al. (2020b) implement controllable generation by introducing gradient guidance, which is given by extra time-dependent classifier or domain knowledge. Computationally, an advantage of the proposed approach over the guidance strategy in Song et al. (2020b) is that no backpropagation through a classifier or other estimation is needed. The class centers, the guidance of ST-DDPM, are predicted in a feed-forward manner and could be cached. To further explore the relationship, we separately visualize the normalized guidance during the reverse process. As shown in Figure 11, the learned class centers with weight-scheduling, as a guidance without extra estimation, show a similar trend with the gradient of classifier during the reverse process. Notice that the learned class centers also have better class separation compared with the unstable gradient guidance.

Figure 11: The normalized guidance for different classes: **Left:** our learned class center and weight-scheduling. **Right:** the gradient of classifier.

## 7 CONCLUSION

In this paper, for the diffusion probabilistic models, we delve into the changes of data distribution during the forward process of the Markov chain and explore the class clustering phenomenon. To enable controllable generation, inspired by the class clustering phenomenon, we devise a novel conditional diffusion probabilistic model ST-DDPM by explicitly modeling the class center in the forward and reverse process, and make an elegant modification to the original formulation. The learned class centers also reflect the common characteristics of datapoints from the same class (e.g. template face). Extensive experiments verify the effectiveness and scalability of our method.

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

## A    EXTENDED DERIVATIONS

### A.1    FORWARD PROCESS GENERAL FORMULA DERIVATION

$$\mathbf{x}_t = \sqrt{\alpha_t}\mathbf{x}_{t-1} + \bar{\alpha}_t^{\frac{1}{4}}(1 - \alpha_t^{\frac{1}{4}})\mathbf{u} + \sqrt{\beta_t}\boldsymbol{\epsilon}_t \tag{14}$$

$$= \sqrt{\alpha_t}\left[\sqrt{\alpha_{t-1}}\mathbf{x}_{t-2} + \bar{\alpha}_{t-1}^{\frac{1}{4}}(1 - \alpha_{t-1}^{\frac{1}{4}})\mathbf{u} + \sqrt{\beta_{t-1}}\boldsymbol{\epsilon}_{t-1}\right] + \bar{\alpha}_t^{\frac{1}{4}}(1 - \alpha_t^{\frac{1}{4}})\mathbf{u} + \sqrt{\beta_t}\boldsymbol{\epsilon}_t \tag{15}$$

$$= \sqrt{\alpha_t\alpha_{t-1}}\mathbf{x}_{t-2} + \sqrt{\alpha_t}\bar{\alpha}_{t-1}^{\frac{1}{4}}(1 - \alpha_{t-1}^{\frac{1}{4}})\mathbf{u} + \sqrt{\alpha_t}\sqrt{\beta_{t-1}}\boldsymbol{\epsilon}_{t-1} + \bar{\alpha}_t^{\frac{1}{4}}(1 - \alpha_t^{\frac{1}{4}})\mathbf{u} + \sqrt{\beta_t}\boldsymbol{\epsilon}_t \tag{16}$$

$$= \sqrt{\alpha_t\alpha_{t-1}}\mathbf{x}_{t-2} + \left[\alpha_t^{\frac{1}{4}}(\alpha_t^{\frac{1}{4}}\bar{\alpha}_{t-1}^{\frac{1}{4}})(1 - \alpha_{t-1}^{\frac{1}{4}}) + \bar{\alpha}_t^{\frac{1}{4}}(1 - \alpha_t^{\frac{1}{4}})\right]\mathbf{u} + \sqrt{\alpha_t\beta_{t-1} + \beta_t}\boldsymbol{\epsilon} \tag{17}$$

$$= \sqrt{\alpha_t\alpha_{t-1}}\mathbf{x}_{t-2} + \bar{\alpha}_t^{\frac{1}{4}}(\alpha_t^{\frac{1}{4}} - \alpha_t^{\frac{1}{4}}\alpha_{t-1}^{\frac{1}{4}} + 1 - \alpha_t^{\frac{1}{4}})\mathbf{u} + \sqrt{\alpha_t(1 - \alpha_{t-1}) + 1 - \alpha_t}\boldsymbol{\epsilon} \tag{18}$$

$$= \sqrt{\alpha_t\alpha_{t-1}}\mathbf{x}_{t-2} + \bar{\alpha}_t^{\frac{1}{4}}(1 - \alpha_t^{\frac{1}{4}}\alpha_{t-1}^{\frac{1}{4}})\mathbf{u} + \sqrt{1 - \alpha_t\alpha_{t-1}}\boldsymbol{\epsilon} \tag{19}$$

$$= \cdots \tag{20}$$

$$= \sqrt{\bar{\alpha}_t}\mathbf{x}_0 + \bar{\alpha}_t^{\frac{1}{4}}(1 - \bar{\alpha}_t^{\frac{1}{4}})\mathbf{u} + \sqrt{1 - \bar{\alpha}_t}\boldsymbol{\epsilon}\,. \tag{21}$$

### A.2    FORWARD PROCESS POSTERIOR DISTRIBUTION DERIVATION

$$q(\mathbf{x}_{t-1}|\mathbf{x}_t, \mathbf{x}_0, \mathbf{u}) = \frac{q(\mathbf{x}_t|\mathbf{x}_{t-1}, \mathbf{x}_0, \mathbf{u})q(\mathbf{x}_{t-1}|\mathbf{x}_0, \mathbf{u})}{q(\mathbf{x}_t|\mathbf{x}_0, \mathbf{u})} \tag{22}$$

$$q(\mathbf{x}_t|\mathbf{x}_{t-1}, \mathbf{x}_0, \mathbf{u}) = q(\mathbf{x}_t|\mathbf{x}_{t-1}, \mathbf{u}) = \mathcal{N}(\mathbf{x}_t; \sqrt{\alpha_t}\mathbf{x}_{t-1} + \bar{\alpha}_t^{\frac{1}{4}}(1 - \alpha_t^{\frac{1}{4}})\mathbf{u}, \beta_t\mathbf{I}) \tag{23}$$

$$q(\mathbf{x}_{t-1}|\mathbf{x}_0, \mathbf{u}) = \mathcal{N}(\mathbf{x}_{t-1}; \sqrt{\bar{\alpha}_{t-1}}\mathbf{x}_0 + \bar{\alpha}_{t-1}^{\frac{1}{4}}(1 - \bar{\alpha}_{t-1}^{\frac{1}{4}})\mathbf{u}, (1 - \bar{\alpha}_{t-1})\mathbf{I}) \tag{24}$$

$$q(\mathbf{x}_t|\mathbf{x}_0, \mathbf{u}) = \mathcal{N}(\mathbf{x}_t; \sqrt{\bar{\alpha}_t}\mathbf{x}_0 + \bar{\alpha}_t^{\frac{1}{4}}(1 - \bar{\alpha}_t^{\frac{1}{4}})\mathbf{u}, (1 - \bar{\alpha}_t)\mathbf{I}) \tag{25}$$

For the reason that all these multivariate Gaussian distributions have diagonal covariance matrix, we can treat $\mathbf{x}$ and $\mathbf{u}$ as univariable, i.e., each element of $\mathbf{x}$ follows following derivation. By denoting $\bar{\alpha}_t^{\frac{1}{4}}(1 - \alpha_t^{\frac{1}{4}})\mathbf{u} = \mathbf{m}_t$ and $\bar{\alpha}_t^{\frac{1}{4}}(1 - \bar{\alpha}_t^{\frac{1}{4}})\mathbf{u} = \mathbf{n}_t$, we can derive $q(\mathbf{x}_{t-1}|\mathbf{x}_t, \mathbf{x}_0, \mathbf{u})$ as follow:

$$q(\mathbf{x}_{t-1}|\mathbf{x}_t, \mathbf{x}_0, \mathbf{u}) = \frac{\frac{1}{\sqrt{2\pi\beta_t}}\exp\left[-\frac{(\mathbf{x}_t - \sqrt{\alpha_t}\mathbf{x}_{t-1} - \mathbf{m}_t)^2}{2\beta_t}\right] \cdot \frac{1}{\sqrt{2\pi(1-\bar{\alpha}_{t-1})}}\exp\left[-\frac{(\mathbf{x}_{t-1} - \sqrt{\bar{\alpha}_{t-1}}\mathbf{x}_0 - \mathbf{n}_{t-1})^2}{2(1-\bar{\alpha}_{t-1})}\right]}{\frac{1}{\sqrt{2\pi(1-\bar{\alpha}_t)}}\exp\left[-\frac{(\mathbf{x}_t - \sqrt{\bar{\alpha}_t}\mathbf{x}_0 - \mathbf{n}_t)^2}{2(1-\bar{\alpha}_t)}\right]} \tag{26}$$

$$= \frac{1}{\sqrt{2\pi}\sqrt{\frac{\beta_t(1-\bar{\alpha}_{t-1})}{1-\bar{\alpha}_t}}}\exp\left[\frac{-E}{2\frac{\beta_t(1-\bar{\alpha}_{t-1})}{(1-\bar{\alpha}_t)}}\right] \tag{27}$$

$$E = \frac{1 - \bar{\alpha}_{t-1}}{1 - \bar{\alpha}_t}\left[\mathbf{x}_t^2 + \alpha_t\mathbf{x}_{t-1}^2 + \mathbf{m}_t^2 + 2\sqrt{\alpha_t}\mathbf{x}_{t-1}\mathbf{m}_t - 2\sqrt{\alpha_t}\mathbf{x}_{t-1}\mathbf{x}_t - 2\mathbf{m}_t\mathbf{x}_t\right] \tag{28}$$

$$+ \frac{\beta_t}{1 - \bar{\alpha}_t}\left[\mathbf{x}_{t-1}^2 + \bar{\alpha}_{t-1}\mathbf{x}_0^2 + \mathbf{n}_{t-1}^2 + 2\sqrt{\bar{\alpha}_{t-1}}\mathbf{x}_0\mathbf{n}_{t-1} - 2\sqrt{\bar{\alpha}_{t-1}}\mathbf{x}_0\mathbf{x}_{t-1} - 2\mathbf{n}_{t-1}\mathbf{x}_{t-1}\right] \tag{29}$$

$$- \frac{\beta_t(1 - \bar{\alpha}_{t-1})}{(1 - \bar{\alpha}_t)^2}\left[\mathbf{x}_t^2 + \bar{\alpha}_t\mathbf{x}_0^2 + \mathbf{n}_t^2 + 2\sqrt{\bar{\alpha}_t}\mathbf{x}_0\mathbf{n}_t - 2\sqrt{\bar{\alpha}_t}\mathbf{x}_0\mathbf{x}_t - 2\mathbf{n}_t\mathbf{x}_t\right] \tag{30}$$

The coefficient of $\mathbf{x}_{t-1}^2$ as $\frac{1-\bar{\alpha}_{t-1}}{1-\bar{\alpha}_t}\alpha_t + \frac{\beta_t}{1-\bar{\alpha}_t} = \frac{\alpha_t - \bar{\alpha}_t + 1 - \alpha_t}{1-\bar{\alpha}_t} = 1$, so we try to transform $E$ to the form of $[\mathbf{x}_{t-1} + (a\mathbf{x}_t + b\mathbf{x}_0 + c)]^2$.

We extract the coefficient of $\mathbf{x}_t^2$ as $a^2$:

$$a^2 = \frac{1 - \bar{\alpha}_{t-1}}{1 - \bar{\alpha}_t} - \frac{\beta_t(1 - \bar{\alpha}_{t-1})}{(1 - \bar{\alpha}_t)^2} \tag{31}$$

$$= \frac{(1 - \bar{\alpha}_{t-1})(1 - \bar{\alpha}_t) - \beta_t(1 - \bar{\alpha}_{t-1})}{(1 - \bar{\alpha}_t)^2} \tag{32}$$

$$= \frac{(1 - \bar{\alpha}_{t-1})(1 - \bar{\alpha}_t - 1 + \alpha_t)}{(1 - \bar{\alpha}_t)^2} \tag{33}$$

$$= \frac{(1 - \bar{\alpha}_{t-1})\alpha_t(1 - \bar{\alpha}_{t-1})}{(1 - \bar{\alpha}_t)^2} \tag{34}$$

$$= \frac{\alpha_t(1 - \bar{\alpha}_{t-1})^2}{(1 - \bar{\alpha}_t)^2} \tag{35}$$

$$a = \pm\frac{\sqrt{\alpha_t}(1 - \bar{\alpha}_{t-1})}{1 - \bar{\alpha}_t} \tag{36}$$

We extract the coefficient of $\mathbf{x}_0^2$ as $b^2$:

$$b^2 = \frac{\beta_t\bar{\alpha}_{t-1}}{1 - \bar{\alpha}_t} - \frac{\beta_t(1 - \bar{\alpha}_{t-1})\bar{\alpha}_t}{(1 - \bar{\alpha}_t)^2} \tag{37}$$

$$= \frac{\beta_t\bar{\alpha}_{t-1}(1 - \bar{\alpha}_t) - \beta_t(1 - \bar{\alpha}_{t-1})\bar{\alpha}_t}{(1 - \bar{\alpha}_t)^2} \tag{38}$$

$$= \frac{\beta_t\bar{\alpha}_{t-1}\left[(1 - \bar{\alpha}_t) - (1 - \bar{\alpha}_{t-1})\alpha_t\right]}{(1 - \bar{\alpha}_t)^2} \tag{39}$$

$$= \frac{\beta_t\bar{\alpha}_{t-1}\left[1 - \bar{\alpha}_t - \alpha_t + \bar{\alpha}_t\right]}{(1 - \bar{\alpha}_t)^2} \tag{40}$$

$$= \frac{\beta_t^2\bar{\alpha}_{t-1}}{(1 - \bar{\alpha}_t)^2} \tag{41}$$

$$b = \pm\frac{\beta_t\sqrt{\bar{\alpha}_{t-1}}}{1 - \bar{\alpha}_t} \tag{42}$$

The coefficient of $\mathbf{x}_{t-1}\mathbf{x}_t$ is $-2\frac{\sqrt{\alpha_t}(1-\bar{\alpha}_{t-1})}{1-\bar{\alpha}_t} = 2a$, which means that $a = -\frac{\sqrt{\alpha_t}(1-\bar{\alpha}_{t-1})}{1-\bar{\alpha}_t}$.

The coefficient of $\mathbf{x}_{t-1}\mathbf{x}_0$ is $-2\frac{\beta_t\sqrt{\bar{\alpha}_{t-1}}}{1-\bar{\alpha}_t} = 2b$, which means that $b = -\frac{\beta_t\sqrt{\bar{\alpha}_{t-1}}}{1-\bar{\alpha}_t}$.

The coefficient of $\mathbf{x}_t\mathbf{x}_0$ is $2\frac{\sqrt{\bar{\alpha}_t}\beta_t(1-\bar{\alpha}_{t-1})}{(1-\bar{\alpha}_t)^2} = 2\frac{\sqrt{\alpha_t}\sqrt{\bar{\alpha}_{t-1}}\beta_t(1-\bar{\alpha}_{t-1})}{(1-\bar{\alpha}_t)^2}$, which is equal to $2ab$.

Note that $\mathbf{m}_t = \mathbf{n}_t - \sqrt{\alpha_t}\mathbf{n}_{t-1}$:

$$\mathbf{n}_t - \sqrt{\alpha_t}\mathbf{n}_{t-1} = \bar{\alpha}_t^{\frac{1}{4}}(1 - \bar{\alpha}_t^{\frac{1}{4}})\mathbf{u} - \sqrt{\alpha_t}\bar{\alpha}_{t-1}^{\frac{1}{4}}(1 - \bar{\alpha}_{t-1}^{\frac{1}{4}})\mathbf{u} \tag{43}$$

$$= \bar{\alpha}_t^{\frac{1}{4}}(1 - \bar{\alpha}_t^{\frac{1}{4}})\mathbf{u} - \alpha_t^{\frac{1}{4}}\alpha_t^{\frac{1}{4}}\bar{\alpha}_{t-1}^{\frac{1}{4}}(1 - \bar{\alpha}_{t-1}^{\frac{1}{4}})\mathbf{u} \tag{44}$$

$$= \bar{\alpha}_t^{\frac{1}{4}}(1 - \bar{\alpha}_t^{\frac{1}{4}})\mathbf{u} - \bar{\alpha}_t^{\frac{1}{4}}(\alpha_t^{\frac{1}{4}} - \bar{\alpha}_t^{\frac{1}{4}})\mathbf{u} \tag{45}$$

$$= \bar{\alpha}_t^{\frac{1}{4}}(1 - \bar{\alpha}_t^{\frac{1}{4}} - \alpha_t^{\frac{1}{4}} + \bar{\alpha}_t^{\frac{1}{4}})\mathbf{u} \tag{46}$$

$$= \bar{\alpha}_t^{\frac{1}{4}}(1 - \alpha_t^{\frac{1}{4}})\mathbf{u} \tag{47}$$

$$= \mathbf{m}_t \tag{48}$$

The coefficient of $\mathbf{x}_{t-1}$ should be equal to $2c$:

$$2\sqrt{\alpha_t}\frac{1-\bar{\alpha}_{t-1}}{1-\bar{\alpha}_t}\mathbf{m}_t - 2\frac{\beta_t}{1-\bar{\alpha}_t}\mathbf{n}_{t-1} \tag{49}$$

$$=2\sqrt{\alpha_t}\frac{1-\bar{\alpha}_{t-1}}{1-\bar{\alpha}_t}(\mathbf{n}_t - \sqrt{\alpha_t}\mathbf{n}_{t-1}) - 2\frac{\beta_t}{1-\bar{\alpha}_t}\mathbf{n}_{t-1} \tag{50}$$

$$=2\sqrt{\alpha_t}\frac{1-\bar{\alpha}_{t-1}}{1-\bar{\alpha}_t}\mathbf{n}_t - 2\alpha_t\frac{1-\bar{\alpha}_{t-1}}{1-\bar{\alpha}_t}\mathbf{n}_{t-1} - 2\frac{\beta_t}{1-\bar{\alpha}_t}\mathbf{n}_{t-1} \tag{51}$$

$$=2\sqrt{\alpha_t}\frac{1-\bar{\alpha}_{t-1}}{1-\bar{\alpha}_t}\mathbf{n}_t - 2\left[\frac{\alpha_t(1-\bar{\alpha}_{t-1})+\beta_t}{1-\bar{\alpha}_t}\right]\mathbf{n}_{t-1} \tag{52}$$

$$=2\sqrt{\alpha_t}\frac{1-\bar{\alpha}_{t-1}}{1-\bar{\alpha}_t}\mathbf{n}_t - 2\left[\frac{\alpha_t - \bar{\alpha}_t + 1 - \alpha_t}{1-\bar{\alpha}_t}\right]\mathbf{n}_{t-1} \tag{53}$$

$$=2\sqrt{\alpha_t}\frac{1-\bar{\alpha}_{t-1}}{1-\bar{\alpha}_t}\mathbf{n}_t - 2\mathbf{n}_{t-1} \tag{54}$$

$$=2\left[\sqrt{\alpha_t}\frac{1-\bar{\alpha}_{t-1}}{1-\bar{\alpha}_t}\mathbf{n}_t - \mathbf{n}_{t-1}\right] \tag{55}$$

$$=2c \tag{56}$$

$$c =\sqrt{\alpha_t}\frac{1-\bar{\alpha}_{t-1}}{1-\bar{\alpha}_t}\mathbf{n}_t - \mathbf{n}_{t-1} \tag{57}$$

$$\tag{58}$$

The coefficient of $\mathbf{x}_t$ should be $2ac$:

$$-2\frac{1-\bar{\alpha}_{t-1}}{1-\bar{\alpha}_t}\mathbf{m}_t + 2\frac{\beta_t(1-\bar{\alpha}_{t-1})}{(1-\bar{\alpha}_t)^2}\mathbf{n}_t \tag{59}$$

$$=-2\frac{1-\bar{\alpha}_{t-1}}{1-\bar{\alpha}_t}(\mathbf{n}_t - \sqrt{\alpha_t}\mathbf{n}_{t-1}) + 2\frac{\beta_t(1-\bar{\alpha}_{t-1})}{(1-\bar{\alpha}_t)^2}\mathbf{n}_t \tag{60}$$

$$=2\left[\frac{\beta_t(1-\bar{\alpha}_{t-1})}{(1-\bar{\alpha}_t)^2} - \frac{1-\bar{\alpha}_{t-1}}{1-\bar{\alpha}_t}\right]\mathbf{n}_t + 2\sqrt{\alpha_t}\frac{1-\bar{\alpha}_{t-1}}{1-\bar{\alpha}_t}\mathbf{n}_{t-1} \tag{61}$$

$$=2\left[\frac{\beta_t(1-\bar{\alpha}_{t-1}) - (1-\bar{\alpha}_t)(1-\bar{\alpha}_{t-1})}{(1-\bar{\alpha}_t)^2}\right]\mathbf{n}_t + 2\sqrt{\alpha_t}\frac{1-\bar{\alpha}_{t-1}}{1-\bar{\alpha}_t}\mathbf{n}_{t-1} \tag{62}$$

$$=2\left[\frac{(1-\bar{\alpha}_{t-1})(\beta_t - 1 + \bar{\alpha}_t)}{(1-\bar{\alpha}_t)^2}\right]\mathbf{n}_t + 2\sqrt{\alpha_t}\frac{1-\bar{\alpha}_{t-1}}{1-\bar{\alpha}_t}\mathbf{n}_{t-1} \tag{63}$$

$$=2\left[\frac{(1-\bar{\alpha}_{t-1})(-\alpha_t + \bar{\alpha}_t)}{(1-\bar{\alpha}_t)^2}\right]\mathbf{n}_t + 2\sqrt{\alpha_t}\frac{1-\bar{\alpha}_{t-1}}{1-\bar{\alpha}_t}\mathbf{n}_{t-1} \tag{64}$$

$$=2\left[\frac{(1-\bar{\alpha}_{t-1})\alpha_t(-1 + \bar{\alpha}_{t-1})}{(1-\bar{\alpha}_t)^2}\right]\mathbf{n}_t + 2\sqrt{\alpha_t}\frac{1-\bar{\alpha}_{t-1}}{1-\bar{\alpha}_t}\mathbf{n}_{t-1} \tag{65}$$

$$=2\left[-\alpha_t\frac{(1-\bar{\alpha}_{t-1})^2}{(1-\bar{\alpha}_t)^2}\right]\mathbf{n}_t + 2\sqrt{\alpha_t}\frac{1-\bar{\alpha}_{t-1}}{1-\bar{\alpha}_t}\mathbf{n}_{t-1} \tag{66}$$

$$=-2\left[\sqrt{\alpha_t}\frac{1-\bar{\alpha}_{t-1}}{1-\bar{\alpha}_t}\mathbf{n}_t - \mathbf{n}_{t-1}\right]\frac{\sqrt{\alpha_t}(1-\bar{\alpha}_{t-1})}{1-\bar{\alpha}_t} \tag{67}$$

$$=2ca \tag{68}$$

The coefficient of $\mathbf{x}_0$ should be $2bc$:

$$-2\sqrt{\bar{\alpha}_t}\frac{\beta_t(1-\bar{\alpha}_{t-1})}{(1-\bar{\alpha}_t)^2}\mathbf{n}_t + 2\sqrt{\bar{\alpha}_{t-1}}\frac{\beta_t}{1-\bar{\alpha}_t}\mathbf{n}_{t-1} \tag{69}$$

$$=-2\left[\frac{\beta_t\sqrt{\bar{\alpha}_{t-1}}}{1-\bar{\alpha}_t}\right]\left[\sqrt{\alpha_t}\frac{1-\bar{\alpha}_{t-1}}{1-\bar{\alpha}_t}\mathbf{n}_t - \mathbf{n}_{t-1}\right] \tag{70}$$

$$=2bc \tag{71}$$

The remaining items should be equal to $c^2$:

$$\frac{1 - \bar{\alpha}_{t-1}}{1 - \bar{\alpha}_t} \mathbf{m}_t^2 + \frac{\beta_t}{1 - \bar{\alpha}_t} \mathbf{n}_{t-1}^2 - \frac{\beta_t(1 - \bar{\alpha}_{t-1})}{(1 - \bar{\alpha}_t)^2} \mathbf{n}_t^2 \tag{72}$$

$$= \frac{1 - \bar{\alpha}_{t-1}}{1 - \bar{\alpha}_t} (\mathbf{n}_t - \sqrt{\alpha_t} \mathbf{n}_{t-1})^2 + \frac{\beta_t}{1 - \bar{\alpha}_t} \mathbf{n}_{t-1}^2 - \frac{\beta_t(1 - \bar{\alpha}_{t-1})}{(1 - \bar{\alpha}_t)^2} \mathbf{n}_t^2 \tag{73}$$

$$= \left[ \frac{1 - \bar{\alpha}_{t-1}}{1 - \bar{\alpha}_t} - \frac{\beta_t(1 - \bar{\alpha}_{t-1})}{(1 - \bar{\alpha}_t)^2} \right] \mathbf{n}_t^2 + \left[ \alpha_t \frac{1 - \bar{\alpha}_{t-1}}{1 - \bar{\alpha}_t} + \frac{\beta_t}{1 - \bar{\alpha}_t} \right] \mathbf{n}_{t-1}^2 - 2\sqrt{\alpha_t} \frac{1 - \bar{\alpha}_{t-1}}{1 - \bar{\alpha}_t} \mathbf{n}_t \mathbf{n}_{t-1} \tag{74}$$

$$= \frac{(1 - \bar{\alpha}_t)(1 - \bar{\alpha}_{t-1}) - \beta_t(1 - \bar{\alpha}_{t-1})}{(1 - \bar{\alpha}_t)^2} \mathbf{n}_t^2 + \left[ \frac{\alpha_t - \bar{\alpha}_t + \beta_t}{1 - \bar{\alpha}_t} \right] \mathbf{n}_{t-1}^2 - 2\sqrt{\alpha_t} \frac{1 - \bar{\alpha}_{t-1}}{1 - \bar{\alpha}_t} \mathbf{n}_t \mathbf{n}_{t-1} \tag{75}$$

$$= \frac{(1 - \bar{\alpha}_{t-1})(1 - \bar{\alpha}_t - \beta_t)}{(1 - \bar{\alpha}_t)^2} \mathbf{n}_t^2 + \left[ \frac{\alpha_t - \bar{\alpha}_t + 1 - \alpha_t}{1 - \bar{\alpha}_t} \right] \mathbf{n}_{t-1}^2 - 2\sqrt{\alpha_t} \frac{1 - \bar{\alpha}_{t-1}}{1 - \bar{\alpha}_t} \mathbf{n}_t \mathbf{n}_{t-1} \tag{76}$$

$$= \frac{(1 - \bar{\alpha}_{t-1})(1 - \bar{\alpha}_t - 1 + \alpha_t)}{(1 - \bar{\alpha}_t)^2} \mathbf{n}_t^2 + \mathbf{n}_{t-1}^2 - 2\sqrt{\alpha_t} \frac{1 - \bar{\alpha}_{t-1}}{1 - \bar{\alpha}_t} \mathbf{n}_t \mathbf{n}_{t-1} \tag{77}$$

$$= \frac{(1 - \bar{\alpha}_{t-1})\alpha_t(-\bar{\alpha}_{t-1} + 1)}{(1 - \bar{\alpha}_t)^2} \mathbf{n}_t^2 + \mathbf{n}_{t-1}^2 - 2\sqrt{\alpha_t} \frac{1 - \bar{\alpha}_{t-1}}{1 - \bar{\alpha}_t} \mathbf{n}_t \mathbf{n}_{t-1} \tag{78}$$

$$= \alpha_t \frac{(1 - \bar{\alpha}_{t-1})^2}{(1 - \bar{\alpha}_t)^2} \mathbf{n}_t^2 + \mathbf{n}_{t-1}^2 - 2\sqrt{\alpha_t} \frac{1 - \bar{\alpha}_{t-1}}{1 - \bar{\alpha}_t} \mathbf{n}_t \mathbf{n}_{t-1} \tag{79}$$

$$= \left[ \sqrt{\alpha_t} \frac{1 - \bar{\alpha}_{t-1}}{1 - \bar{\alpha}_t} \mathbf{n}_t - \mathbf{n}_{t-1} \right]^2 \tag{80}$$

$$= c^2 \tag{81}$$

Finally we have $a = -\frac{\sqrt{\alpha_t}(1 - \bar{\alpha}_{t-1})}{1 - \bar{\alpha}_t}$, $b = -\frac{\beta_t \sqrt{\bar{\alpha}_{t-1}}}{1 - \bar{\alpha}_t}$, $c = \sqrt{\alpha_t} \frac{1 - \bar{\alpha}_{t-1}}{1 - \bar{\alpha}_t} \mathbf{n}_t - \mathbf{n}_{t-1}$ and

$$E = [\mathbf{x}_{t-1} + (a\mathbf{x}_t + b\mathbf{x}_0 + c)]^2 \tag{82}$$

$$= \left[ \mathbf{x}_{t-1} + (-\frac{\sqrt{\alpha_t}(1 - \bar{\alpha}_{t-1})}{1 - \bar{\alpha}_t} \mathbf{x}_t - \frac{\beta_t \sqrt{\bar{\alpha}_{t-1}}}{1 - \bar{\alpha}_t} \mathbf{x}_0 + \sqrt{\alpha_t} \frac{1 - \bar{\alpha}_{t-1}}{1 - \bar{\alpha}_t} \mathbf{n}_t - \mathbf{n}_{t-1}) \right]^2 \tag{83}$$

Now we have the forward process posterior distribution as follows:

$$q(\mathbf{x}_{t-1}|\mathbf{x}_t, \mathbf{x}_0, \mathbf{u}) = \frac{1}{\sqrt{2\pi} \sqrt{\frac{\beta_t(1 - \bar{\alpha}_{t-1})}{1 - \bar{\alpha}_t}}} \exp\left[ \frac{-E}{2\frac{\beta_t(1 - \bar{\alpha}_{t-1})}{(1 - \bar{\alpha}_t)}} \right] = \frac{1}{\sqrt{2\pi}\sigma} \exp\left[ \frac{-(\mathbf{x}_{t-1} - \mu)^2}{2\sigma^2} \right] \tag{84}$$

which is right a probability density function of some Gaussian distribution. Then we have:

$$\sigma^2 = \frac{\beta_t(1 - \bar{\alpha}_{t-1})}{(1 - \bar{\alpha}_t)} \tag{85}$$

$$\mu = \frac{\sqrt{\alpha_t}(1 - \bar{\alpha}_{t-1})}{1 - \bar{\alpha}_t}\mathbf{x}_t + \frac{\beta_t\sqrt{\bar{\alpha}_{t-1}}}{1 - \bar{\alpha}_t}\mathbf{x}_0 - \sqrt{\alpha_t}\frac{1 - \bar{\alpha}_{t-1}}{1 - \bar{\alpha}_t}\mathbf{n}_t + \mathbf{n}_{t-1} \tag{86}$$

$$= \frac{\sqrt{\alpha_t}(1 - \bar{\alpha}_{t-1})}{1 - \bar{\alpha}_t}\mathbf{x}_t + \frac{\beta_t\sqrt{\bar{\alpha}_{t-1}}}{1 - \bar{\alpha}_t}\frac{\mathbf{x}_t - \mathbf{n}_t - \sqrt{1 - \bar{\alpha}_t}\boldsymbol{\epsilon}_t}{\sqrt{\bar{\alpha}_t}} - \sqrt{\alpha_t}\frac{1 - \bar{\alpha}_{t-1}}{1 - \bar{\alpha}_t}\mathbf{n}_t + \mathbf{n}_{t-1} \tag{87}$$

$$= \frac{\sqrt{\alpha_t}(1 - \bar{\alpha}_{t-1})}{1 - \bar{\alpha}_t}\mathbf{x}_t + \frac{\beta_t}{1 - \bar{\alpha}_t}\frac{\mathbf{x}_t - \mathbf{n}_t - \sqrt{1 - \bar{\alpha}_t}\boldsymbol{\epsilon}}{\sqrt{\alpha_t}} - \sqrt{\alpha_t}\frac{1 - \bar{\alpha}_{t-1}}{1 - \bar{\alpha}_t}\mathbf{n}_t + \mathbf{n}_{t-1} \tag{88}$$

$$= \left[\frac{\alpha_t(1 - \bar{\alpha}_{t-1})}{(1 - \bar{\alpha}_t)\sqrt{\alpha_t}} + \frac{\beta_t}{(1 - \bar{\alpha}_t)\sqrt{\alpha_t}}\right]\mathbf{x}_t - \frac{\beta_t\sqrt{1 - \bar{\alpha}_t}}{\sqrt{\alpha_t}(1 - \bar{\alpha}_t)}\boldsymbol{\epsilon} - \left[\frac{\beta_t}{(1 - \bar{\alpha}_t)\sqrt{\alpha_t}} + \frac{\alpha_t(1 - \bar{\alpha}_{t-1})}{(1 - \bar{\alpha}_t)\sqrt{\alpha_t}}\right]\mathbf{n}_t + \mathbf{n}_{t-1} \tag{89}$$

$$= \frac{\alpha_t - \bar{\alpha}_t + \beta_t}{(1 - \bar{\alpha}_t)\sqrt{\alpha_t}}\mathbf{x}_t - \frac{\beta_t}{\sqrt{\alpha_t}\sqrt{1 - \bar{\alpha}_t}}\boldsymbol{\epsilon} - \left[\frac{\beta_t + \alpha_t - \bar{\alpha}_t}{(1 - \bar{\alpha}_t)\sqrt{\alpha_t}}\right]\mathbf{n}_t + \mathbf{n}_{t-1} \tag{90}$$

$$= \frac{1}{\sqrt{\alpha_t}}\mathbf{x}_t - \frac{\beta_t}{\sqrt{\alpha_t}\sqrt{1 - \bar{\alpha}_t}}\boldsymbol{\epsilon} - \frac{1}{\sqrt{\alpha_t}}\mathbf{n}_t + \mathbf{n}_{t-1} \tag{91}$$

$$= \frac{1}{\sqrt{\alpha_t}}\mathbf{x}_t - \frac{\beta_t}{\sqrt{\alpha_t}\sqrt{1 - \bar{\alpha}_t}}\boldsymbol{\epsilon} - \frac{1}{\sqrt{\alpha_t}}(\mathbf{n}_t - \sqrt{\alpha_t}\mathbf{n}_{t-1}) \tag{92}$$

$$= \frac{1}{\sqrt{\alpha_t}}\mathbf{x}_t - \frac{\beta_t}{\sqrt{\alpha_t}\sqrt{1 - \bar{\alpha}_t}}\boldsymbol{\epsilon} - \frac{1}{\sqrt{\alpha_t}}\mathbf{m}_t \tag{93}$$

$$= \frac{1}{\sqrt{\alpha_t}}\left(\mathbf{x}_t - \frac{\beta_t}{\sqrt{1 - \bar{\alpha}_t}}\boldsymbol{\epsilon} - \mathbf{m}_t\right) \tag{94}$$

## B  IMPLEMENTATION DETAILS

Follows the backbone of PixelCNN++ Salimans et al. (2017) and DDPM Ho et al. (2020), our employ a U-Net architecture based on a Wide ResNet for noise prediction $\boldsymbol{\epsilon}_\theta$ and replace the weight normalization Salimans & Kingma (2016) with a group normalization Wu & He (2018) to make the implementation simpler. We use four feature map resolutions for 32×32 models, and six resolutions for 64×64 and 256×256 models. All models have two convolutional residual blocks per resolution level and self-attention blocks at 16×16 resolution between convolutional blocks. The time embedding is then specified by adding the Transformer Vaswani et al. (2017) sinusoidal position embedding into each residual block.

To predict the class centers $\boldsymbol{\mu}_\phi$, we stack convolutional residual blocks and upsample layers to map feature vectors to three-channels map for CIFAR-10 and LFW dataset. For image inpainting on CelebA-HQ, LSUN-church, Place2 datasets, we also employ a U-Net architecture for pixel-to-pixel prediction. For text-to-image synthesis on CUB bird dataset, we stack convolutional residual blocks and upsample layers with attention mechanism from the pre-trained word embeddings to predict the class center.

For image inpainting, we use the Irregular Mask Dataset collected by Liu et al. (2018), which contains 55,116 irregular raw masks for training and 24,866 for testing. During training, for each image in the batch, we first randomly sample a mask from 55,116 training masks, then perform some random augmentations on the mask, finally we use it to mask the image and get our class center for training. So the training masks are different all the time. The mask is irregular and may be 100During testing, we use 12,000 test masks sampled and augmented from 24,866 raw testing masks. These 12,000 masks are categorized by hole size according to hole-to-image area ratios (0-20%, 20-40%, 40-60%).

Follow the training setting of DDPM Ho et al. (2020), we set $T = 1000$ and set the diffusion rate increasing linearly from $\beta_1 = 10^{-4}$ to $\beta_T = 0.02$ for all experiments. The dropout rate is set to 0.1. We use the Adam optimizer, and set the learning rate to $2 \times 10^{-4}$ for 32×32 images and $2 \times 10^{-5}$ for higher resolution without any sweeping. The batch size is set to 128 for CIFAR-10, 64 for LFW, and 24 for high-resolution Places2, LSUN-church, CUB-bird and Celea-HQ datasets. We use EMA on model parameters with a decay factor of 0.9999. We train the 32×32 models with four Nvidia RTX 2080Ti GPUs and higher resolution models with four A6000 GPUs.

Table 2: CIFAR-10 results. NLL measured in bits/dim.

| Model | FID↓ | NLL Test (Train) |
|---|---|---|
| **Unconditional** | | |
| Gated PixelCNN | 65.93 | 3.03 (2.90) |
| DDPM | 3.17 | ≤3.75 (3.72) |
| **Conditional** | | |
| cond. DDPM | 3.82 | ≤3.75 (3.72) |
| ST-DDPM | 3.05 | ≤3.74 (3.69) |

Table 3: Evaluation results on the Place2 dataset.

| | FID↓ | | |
|---|---|---|---|
| Mask | 0-20% | 20-40% | 40-60% |
| CA Yu et al. (2018) | 4.8586 | 18.4190 | 37.9432 |
| EdgeConnect Nazeri et al. (2019) | 3.0097 | 7.2635 | 19.003 |
| StructureFlow Ren et al. (2019) | 2.9420 | 7.0354 | 22.3803 |
| cond. DDPM | 2.0401 | 6.5726 | 17.2834 |
| ST-DDPM | **1.7498** | **6.1887** | **14.5111** |

For comparison, following Song et al. (2020b); Dhariwal & Nichol (2021), we introduce a time-dependent classifier and compute the gradient for class-conditional sampling (grad. DDPM), given by:

$$\boldsymbol{\epsilon} = \boldsymbol{\epsilon}_\theta(\boldsymbol{x}_t, t) - \sqrt{1 - \bar{\alpha}_t} \nabla_{\boldsymbol{x}_t} \log p_\phi(y|\boldsymbol{x}_t) \tag{95}$$

where $\phi$ is the parameters of the pretrained classifier, $y$ is the class label. Also, we build the simple conditional DDPM (cond. DDPM) by directly injecting a class embedding along with the timestep embedding into the denoising network, given by:

$$\boldsymbol{\epsilon} = \boldsymbol{\epsilon}_\theta(\boldsymbol{x}_t, \boldsymbol{c}, t) \tag{96}$$

where $\boldsymbol{c}$ is the given condition (e.g. class label).

## C  MORE EVALUATION RESULTS

We measure the negative log likelihoods (lossless codelengths) on CIFAR-10 dataset, given in Table 2.

Further, to evaluate the model on image inpainting, we follow prior works Yu et al. (2019); Liu et al. (2018) by reporting the FID score on Place2 dataset. Table 3 shows the comparison results. The ST-DDPM achieves competitive results comparable to prior GAN-based methods for the FID score, showing the great potential of our method.

Table 4: Evaluation results of text-to-image generation.

| Model | FID↓ | IS↑ |
|---|---|---|
| StackGAN (Zhang et al. (2017)) | 51.89 | 3.70 |
| StackGAN++ (Zhang et al. (2018)) | 15.30 | 3.82 |
| FusedGAN (Bodla et al. (2018)) | - | 3.92 |
| AttnGAN (Xu et al. (2018)) | - | 4.36 |
| MirrorGAN (Qiao et al. (2019)) | - | 4.56 |
| AGAN-CL (Wang et al. (2020)) | - | 4.97 |
| ST-DDPM | 18.18 | 4.52 |

## D  FASTER SAMPLING WITH STRIDED SAMPLING

Given the Eq.(7), we can get a transfer relation as follows:

$$\mathbf{x}_t = \sqrt{\bar{\alpha}_t}\mathbf{x}_0 + \mathbf{n}_t + \sqrt{\beta_t}\boldsymbol{\epsilon}_t \tag{97}$$

As $t$ increases, $\sqrt{\bar{\alpha}_t}$ get closer to zero, leading to an approximate transfer relation as follows:

$$\mathbf{x}_t \approx \mathbf{n}_t + \sqrt{\beta_t}\boldsymbol{\epsilon}_t \tag{98}$$

Therefore, we provide another thought for faster sampling that is to select a earlier starting timestep.

Also, inspired by DDIM Song et al. (2020a) with strided sampling, we adapt our formulations into similar non-Markovian inference processes, which enables new and fast generative processes. Specifically, we introduce a family of inference distribution indexed by $\sigma \in \mathbb{R}^T_{\geq 0}$:

$$q_\sigma(\mathbf{x}_{1:T}|\mathbf{x}_0) = q_\sigma(\mathbf{x}_T|\mathbf{x}_0)\prod_{t=2}^{T} q_\sigma(\mathbf{x}_{t-1}|\mathbf{x}_t, \mathbf{x}_0) \tag{99}$$

where

$$q_\sigma(\mathbf{x}_T|\mathbf{x}_0) = \mathcal{N}(\mathbf{x}_T; \sqrt{\bar{\alpha}_T}\mathbf{x}_0 + \mathbf{n}_T, (1-\bar{\alpha}_T)\mathbf{I}) \tag{100}$$

and for all $t > 1$

$$q_\sigma(\mathbf{x}_{t-1}|\mathbf{x}_t, \mathbf{x}_0) = \mathcal{N}(\mathbf{x}_{t-1}; \sqrt{\bar{\alpha}_{t-1}}\mathbf{x}_0 + \mathbf{n}_{t-1} - \sqrt{1 - \bar{\alpha}_{t-1} - \sigma_t^2} \cdot \frac{\mathbf{x}_t - \sqrt{\bar{\alpha}_t}\mathbf{x}_0 - \mathbf{n}_t}{\sqrt{1-\bar{\alpha}_t}}, \sigma_t^2\mathbf{I}). \tag{101}$$

They are designed to ensure that $q_\sigma(\mathbf{x}_t|\mathbf{x}_0) = \mathcal{N}(\mathbf{x}_t; \sqrt{\bar{\alpha}_t}\mathbf{x}_0 + \mathbf{n}_t, (1-\bar{\alpha}_t)\mathbf{I})$ for all $t$.

Then we employ a trainable generative process $p_\theta(\mathbf{x}_{0:T})$ where each $p_\theta(\mathbf{x}_{t-1}|\mathbf{x}_t)$ models the reverse conditional distribution $q_\sigma(\mathbf{x}_{t-1}|\mathbf{x}_t, \mathbf{x}_0)$. Specifically, for some $x_0 \sim q(x_0)$ and $\epsilon_t \sim \mathcal{N}(\mathbf{0}, \mathbf{I})$, we can obtain $x_t$ using Eq. (7) and train a model $\epsilon_\theta$ to predict $\epsilon_t$ from $x_t$ without $x_0$. We can then predict the denoised observation $f_\theta(\mathbf{x}_t, t)$ with $\hat{\epsilon}_t$:

$$\hat{\epsilon}_t = \epsilon_\theta(\mathbf{x}_t, t) - \frac{\mathbf{n}_t}{\sqrt{1-\bar{\alpha}_t}}, \tag{102}$$

$$f_\theta(\mathbf{x}_t, t) = \frac{\mathbf{x}_t - \mathbf{n}_t - \sqrt{1-\bar{\alpha}_t} \cdot \hat{\epsilon}_t}{\sqrt{\bar{\alpha}_t}}. \tag{103}$$

Then we can define the generative process with a fixed prior $p_\theta(\mathbf{x}_T) = \mathcal{N}(\mathbf{x}_T; \mathbf{0}, \mathbf{I})$ and

$$p_\theta(\mathbf{x}_{t-1}|\mathbf{x}_t) = \begin{cases} \mathcal{N}(\mathbf{x}_0; f_\theta(\mathbf{x}_1, 1), \sigma_1^2\mathbf{I}) & \text{if t=1} \\ q_\sigma(\mathbf{x}_{t-1}|\mathbf{x}_t, f_\theta(\mathbf{x}_t, t)) & \text{otherwise,} \end{cases} \tag{104}$$

from which one can generate a sample $\mathbf{x}_{t-1}$ from $\mathbf{x}_t$ via:

$$\mathbf{x}_{t-1} = \sqrt{\bar{\alpha}_{t-1}} \cdot \frac{\mathbf{x}_t - \mathbf{n}_t - \sqrt{1-\bar{\alpha}_t} \cdot \hat{\epsilon}_t}{\sqrt{\bar{\alpha}_t}} + \mathbf{n}_{t-1} + \sqrt{1 - \bar{\alpha}_{t-1} - \sigma_t^2} \cdot \hat{\epsilon}_t + \sigma_t\epsilon_t. \tag{105}$$

We do these modifications to ensure the optimal solution for ST-DDIM is also the same as that for ST-DDPM, which enables us to adopt other forward processes with smaller steps for accelerated sampling without having to retrain the model. We take the same settings with DDIM for $\tau$ and $\sigma$, where $\tau$ is a sub-sequences of $[1, \cdots, T]$ and $\sigma_{\tau_i}(\eta) = \eta\sqrt{(1-\bar{\alpha}_{\tau_{i-1}})(1-\bar{\alpha}_{\tau_i})}\sqrt{1 - \frac{\bar{\alpha}_{\tau_i}}{\bar{\alpha}_{\tau_{i-1}}}}$. $\eta$ is a hyperparameter that we can directly control.

The evaluation results of FID on the CIFAR-10 dataset with strided sampling (SS) and earlier starting points (ES) with different $\eta$ are presented in Table 5. With earlier starting and strided sampling, our method can achieve a speedup of $11.1\times$ without significant decreasing in sample quality. Compared with existing works Watson et al. (2021); Song et al. (2020a), our method can also achieve competitive evaluation results with the same number of steps. Also, more generated examples with different $\eta$ and sampling steps $T$ are presented in Figure 12.

Table 5: Evaluation results of FID on the CIFAR-10 dataset for ST-DDPM with strided sampling (SS) and earlier starting (ES), and the complete sampling step $T = 1000$.

| | Step | SS | | | | | ES + SS | |
| --- | --- | --- | --- | --- | --- | --- | --- | --- |
| | | 10 | 20 | 50 | 100 | 300 | 90 | 270 |
| $\eta$ | 0.0 | 13.69 | 6.42 | 4.64 | 4.29 | 4.09 | 4.45 | 4.10 |
| | 0.2 | 14.84 | 7.54 | 4.89 | 4.68 | 4.18 | 4.62 | 4.33 |
| | 0.5 | 15.40 | 8.88 | 5.81 | 4.88 | 4.25 | 4.88 | 4.53 |
| | 1.0 | 27.67 | 16.23 | 8.88 | 6.27 | 4.54 | 6.25 | 4.61 |

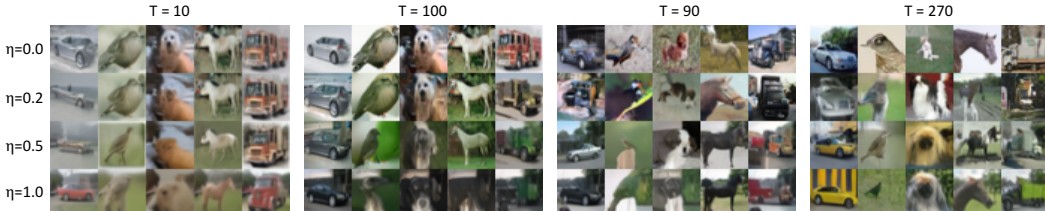

Figure 12: More generated samples with different sampling steps.

# E ADDITIONAL SAMPLES

Figure 13, 15, 16, 17, 18 present more generated examples separately on the CIFAR-10 Krizhevsky et al. (2009), CelebA-HQ Liu et al. (2015), LSUN-church Yu et al. (2015), Place2 Zhou et al. (2017) and LFW Huang et al. (2008) dataset. We also visualize the class center in the second column in Figure 15, 16, 17.

Figure 14 shows the generated examples and class centers on the widely-used 256×256 CUB bird Wah et al. (2011) dataset. Notice that the class centers are also bird-like templates, demonstrating the effectiveness of our method.

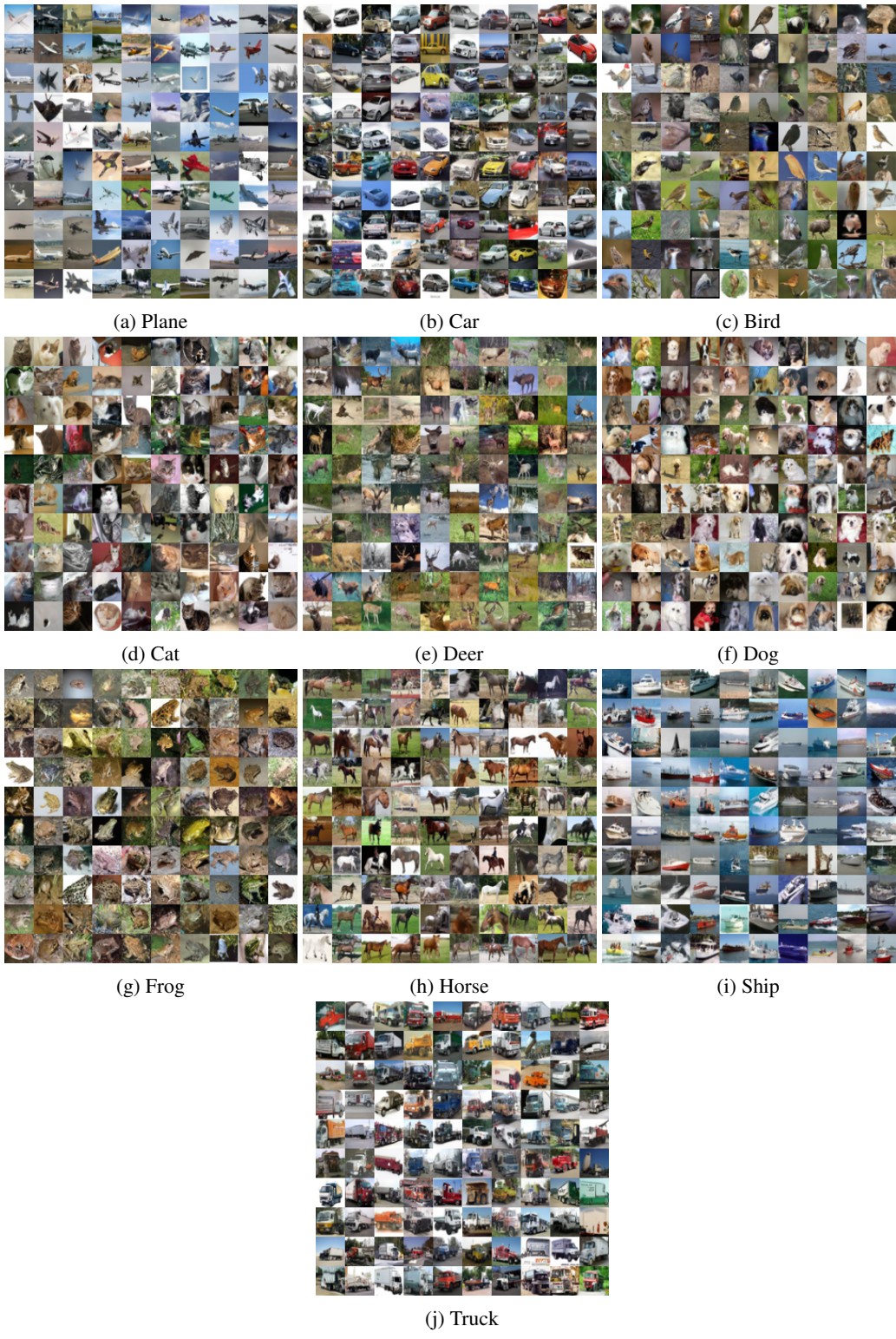

Figure 13: More class-conditional samples on 32×32 CIFAR-10.

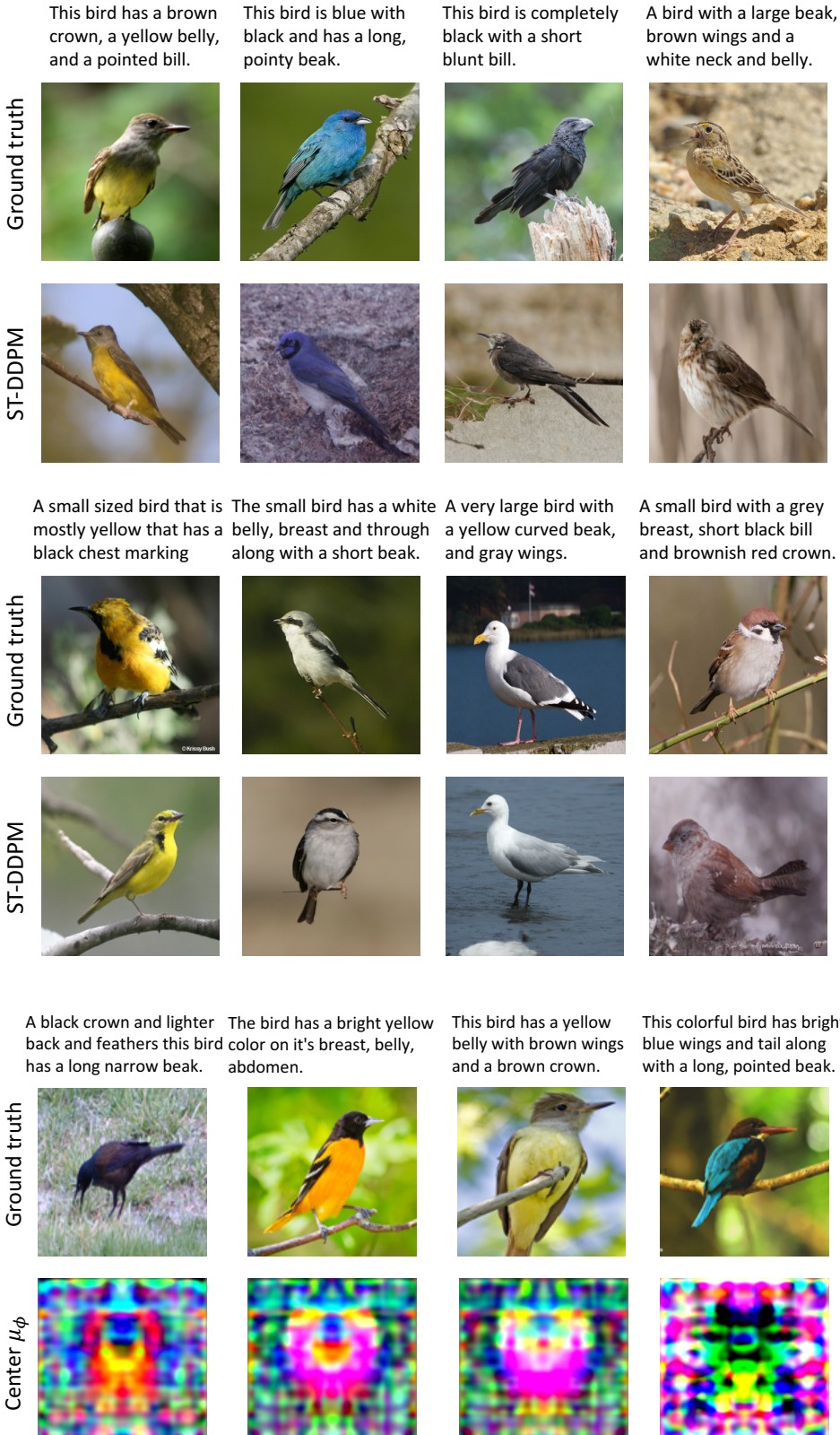

Figure 14: Text-to-image synthesis results on the 256×256 CUB bird dataset.

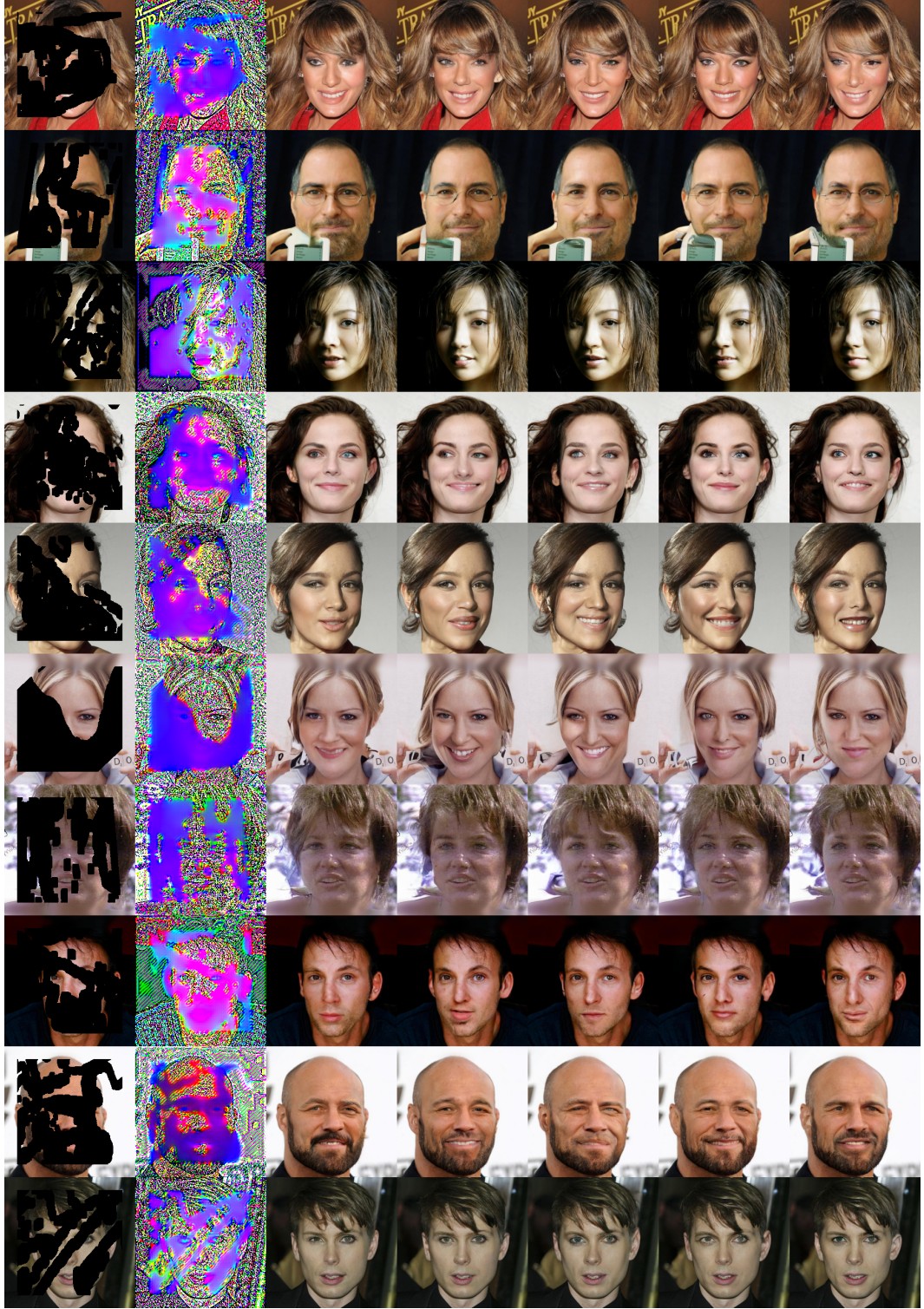

Figure 15: More generated samples on 256×256 CelebA-HQ dataset.

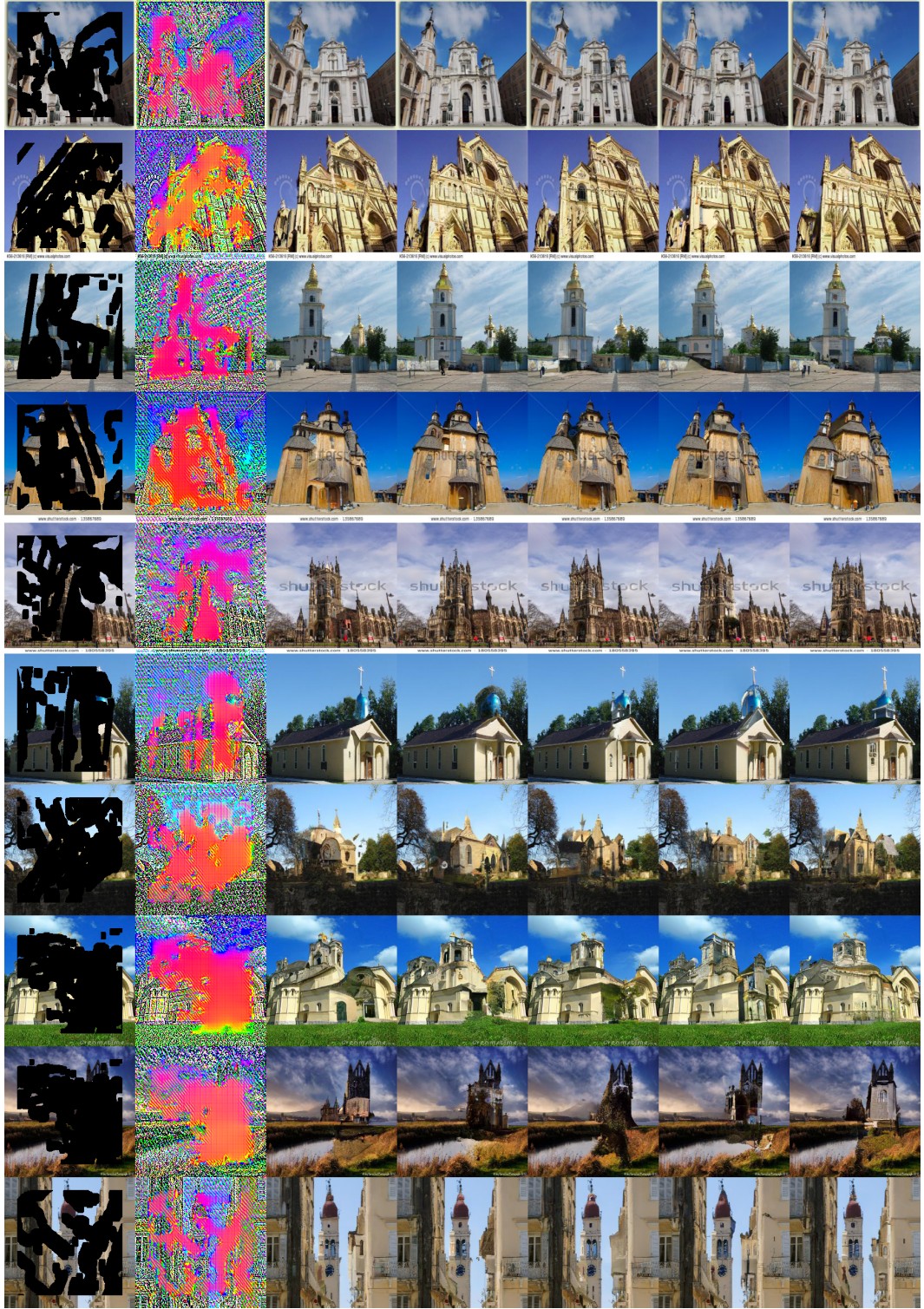

Figure 16: More generated samples on 256×256 LSUN-church dataset.

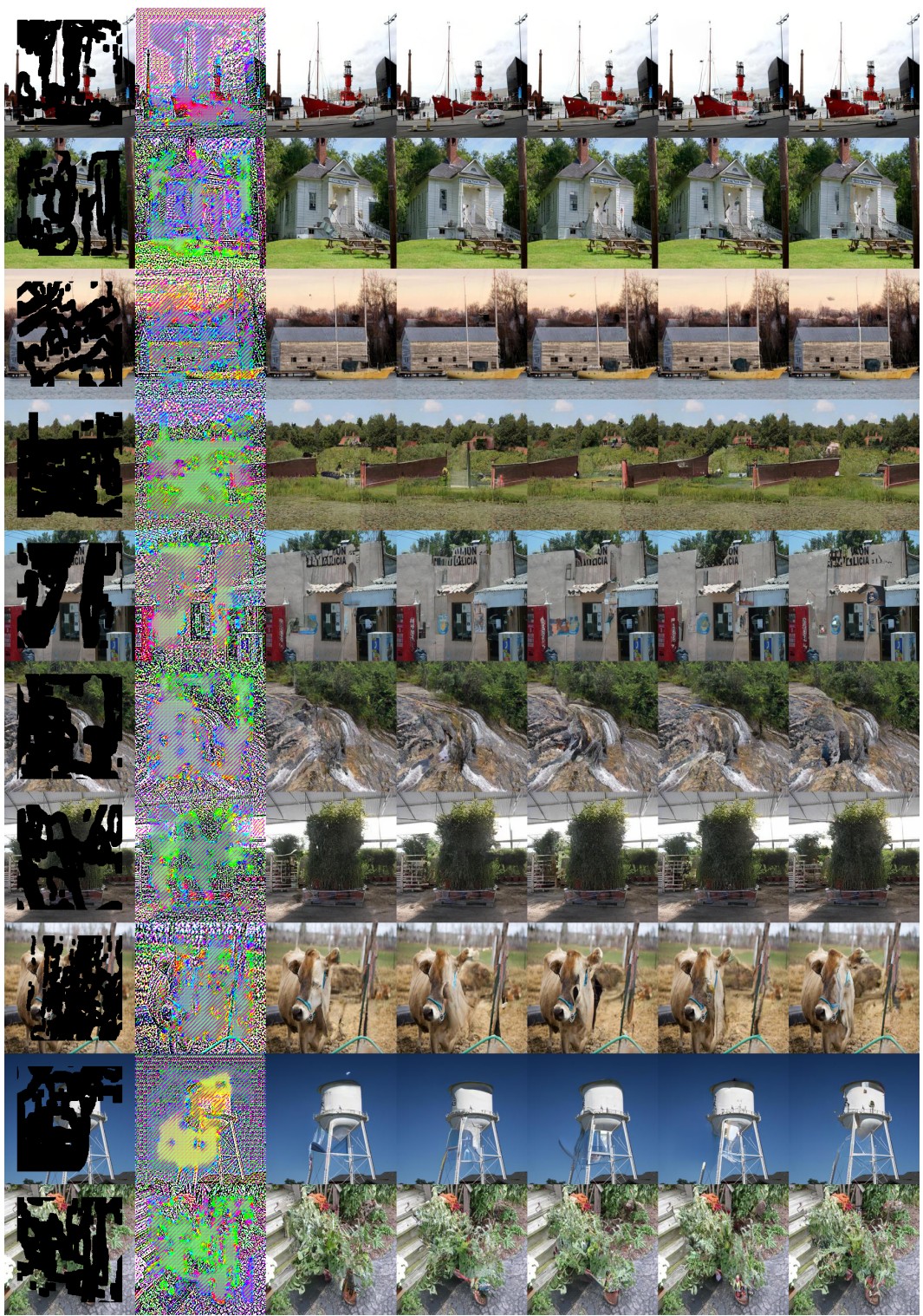

Figure 17: More generated samples on 256×256 Place2 dataset.

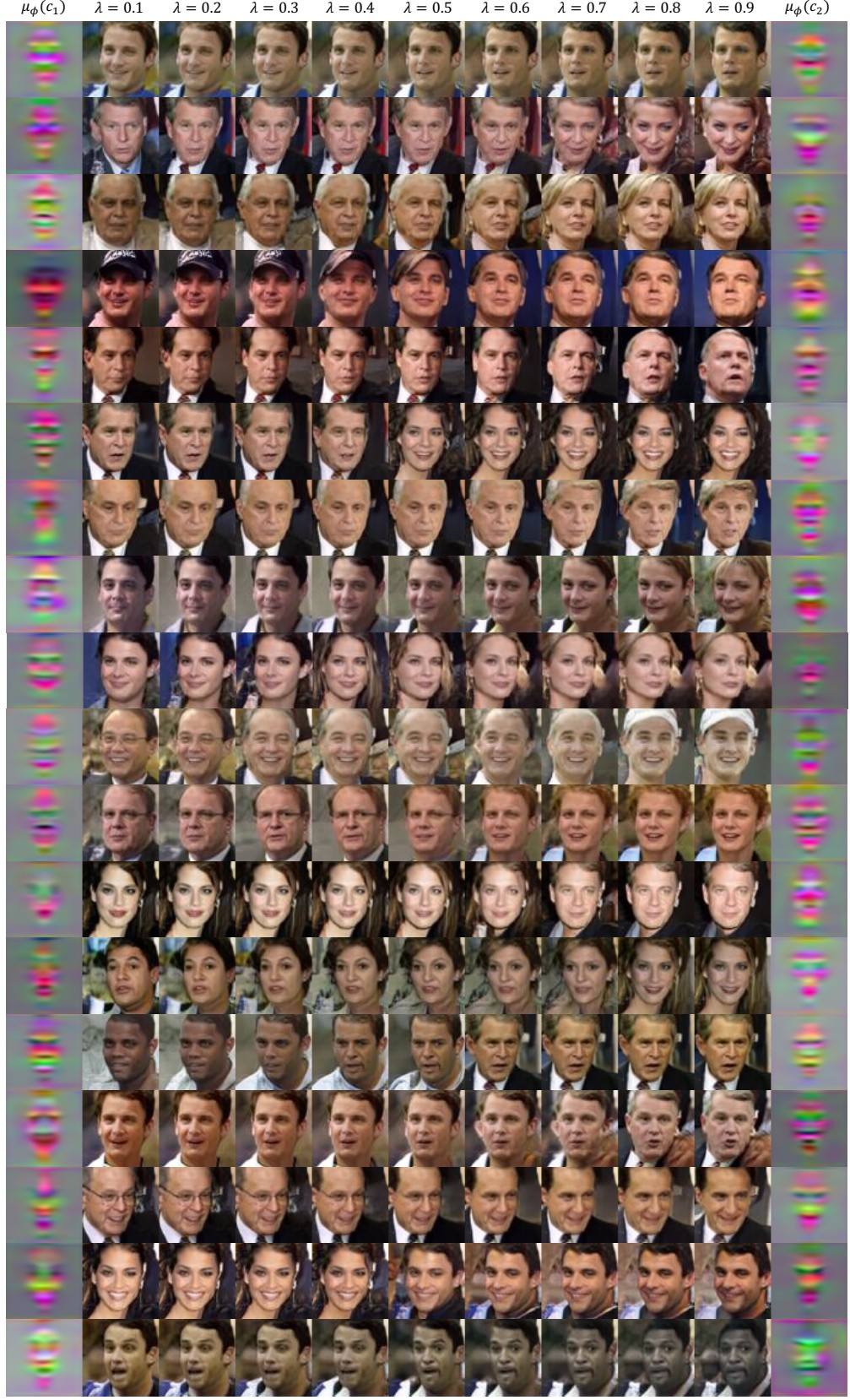

Figure 18: More interpolation samples on 64×64 LFW dataset.

