# OpenReview forum: "ST-DDPM: Explore Class Clustering for Conditional Diffusion Probabilistic Models"
_ICLR.cc/2022/Conference — ICLR 2022 Submitted_

### Official Review · Reviewer_HJtB · 2021-10-27

**Correctness:** 3
**Technical Novelty And Significance:** 2
**Empirical Novelty And Significance:** 2
**Recommendation:** 6
**Confidence:** 4

**Main Review:**

Strengths:
The conditional framework proposed in this paper is general and is applicable to different tasks. The learned condition embedding contains some semantic information.

Questions:
1. The earlier starting (ES) is only slightly mentioned in Sec.6.5. Although the author claims that details can be found in Appendix B, I didn't find how it is derived and what is the principle. Besides, it seems that only using strided sampling (SS) can work even better than combing ES and SS, e.g., Table 4 in Appendix shows that SS with 100 steps can achieve a FID of 4.29, which is better than 4.45 of ES+SS.

2. Is the decoupling of $\epsilon_\theta$ reasonable? According to Eq.(11) and Eq.(12), the optimal $\epsilon_\theta^*(x_t, u, t)=E[\epsilon|x_t]=E[\frac{x_t - \sqrt{\overline{\alpha}_t} x_0 }{\sqrt{1-\overline{\alpha}_t}}|x_t] - \frac{n_t}{\sqrt{1-\overline{\alpha}_t}}$. The first term is still related to $u$, since the joint distribution of $x_0, x_t$ is $q(x_0|c) q(x_t|x_0,c)$, which is related to $u$. However, the author models the first term with $\epsilon_\theta(x_t, t)$, which is independent of $u$.

3. Algorithm 1 and Algorithm 2 don't show how to get the label $c$. Besides, in Algorithm1, shouldn't $u=u_\phi(x_0)$ be $u=u_\phi(c)$?

4. At the beginning of Section 5, the author denotes $u_\phi(c)$ as the class center. However, $u_\phi(c)$ is actually learned, which is not rigorously the class center. It is likely to cause misunderstanding.

5. As mentioned earlier, the optimal $\epsilon_\theta^*(x_t, u, t)=E[\epsilon|x_t]$. What will the optimal condition embedding $u_\phi(c)$ be?

6. I didn't find a strong relationship between the clustering phenomenon and the formulation of ST-DDPM. How the phenomenon motivates ST-DDPM should be discussed in detail and formally.

7. To distinguish this paper from prior conditional methods, the strengths than the traditional conditional DDPM (i.e., the cond. DDPM in Table1) should be discussed more.

8. This paper is about conditional diffusion models, thereby it would be better if experimental comparisons with prior conditional diffusion models (e.g., [1*, 2* ,3*]) are added. The author should at least discuss these works in the related work section.

9. Some missing experimental details. For example, what is the batch size?

[1*] Improved Denoising Diffusion Probabilistic Models

[2*] Diffusion Models Beat GANs on Image Synthesis

[3*] ILVR: Conditioning Method for Denoising Diffusion Probabilistic Models


**Summary Of The Paper:**

The paper proposes a new forward conditional process. The transition kernel is controlled by a condition embedding. It also derives a reverse process. The $\epsilon_\theta$ module is decomposed into two parts. The paper conducts experiments on different conditional generation tasks.

**Summary Of The Review:**

This work has some questions, which should be addressed to reach the acceptance threshold.

---

> ### Author Response · Authors · 2021-11-11
> **Responses to Reviewer HJtB**
>
> #### **1. More details about earlier starting (ES).**
> Given the Eq.(7), we can get a transfer relation as follows:
>
> $
> \mathbf{x}_t = \sqrt{\bar{\alpha}_t}\mathbf{x}_0 + \mathbf{n}_t + \sqrt{\beta_t}\boldsymbol{\epsilon}_t
> $
>
> As $t$ increases, $\sqrt{\bar{\alpha}_t}$ get closer to zero, leading to
> an approximate transfer relation as follows:
>
> $
> \mathbf{x}_t \approx \mathbf{n}_t + \sqrt{\beta_t}\boldsymbol{\epsilon}_t
> $
>
> Therefore, we provide another thought for faster sampling that is to select a earlier starting timestep. And thanks for your advice and we have already added the detailed derivation in the rebuttal revision.
>
> For evaluation results, when $\eta = 0.0$, few steps are taken for sampling. The "ES+SS" achieves similar results(FID = 4.45) with only **90 steps** compared to "SS" with **100 steps**(FID = 4.29). And when $\eta = 0.2, 0.5, 1.0$, "ES+SS" achieves more amazing results, as shown in Table 4 of Appendix D.
>
> #### **2. About the decoupling.**
> Sure it is and $\boldsymbol{\mu}$ could be taken as one of the input.
> However, from another pespective, the first item is a function with the input of $\mathbf{x}_t$ and $\mathbf{x}_0$.
> Intuitively, the first item could be considered as a denoising network.
> With given $\mathbf{x}_t$, the network is able to present a reasonable decoding path and the second item adjusts it. And during training, the denoising network actually absorbs the class information and can tell a reasonable path.
>
> Also, the decoupled item $\mathbf{n}_t$ can be considered as a reasonable gudiance and show a similar trend with the gradient of classifier during the reverse process compared with score-based methods using bayes rule(Section 6.6).
>
> #### **3. About the notations in Algorithm1 and Algorithm2.**
> As you said, $\boldsymbol{\mu} = \boldsymbol{\mu}_\phi(\mathbf{c})$ is the encoded condition, and $c$ is the given condition. (e.g. one-hot label for Cifar-10, query description for CUB-brid, corrupted image for LSUN). In Algorithm1 and Algorithm2, we simply denote the encoded condition as $\boldsymbol{\mu}$.
>
> #### **4. About $\boldsymbol{\mu}_\phi(\mathbf{c})$ in Section 5.**
> The proposed $\boldsymbol{\mu}_\phi(\mathbf{c})$ is given in the forward equation(Eq.(6) and Eq.(7)). The formulation will pull the datapoints towards the $\boldsymbol{\mu}$, making it a center. And as $\bar{\alpha}_t$ gets smaller and closer to $0$, the data distribution is ultimately converted into noise distribution.
>
> Thanks for your advice. We will make it clearer in the revision.
>
> #### **5. About the optimal condition embedding $\boldsymbol{\mu}_\phi(\mathbf{c})$.**
> The optimal $\epsilon^{\star}$ is to minimize the given expectation as follows:
>
> $
> \mathbb{E}_{\mathbf{x}_t,\boldsymbol{\epsilon}}\left[ \parallel \boldsymbol{\epsilon} - \boldsymbol{\epsilon}_\theta \parallel^{2} \right]
> $
>
> And the optimal condition embedding $\boldsymbol{\mu}_\phi(\mathbf{c})$ can be learned through the optimization. Since we explicitly model the item as centers, the qualitative results also show interpretability in Figure 4, 6, 8 and 9.
>
> #### **6. About the connection between the clustering phenomenon and the formulation.**
>
> In the early stage of forward process, the data distribution maintain stable intra-class variance and inter-class variance(Figure 2 and 3), demonstrating stable class separation and class clustering, which motivates us to explicitly introduce class centers to describe the common attribute and model the datapoints from specificity to universality.
>
> And the variance ratio suddenly increases and the data distribution turns into noise, making us devise the strategy of weight-scheduling and model the datapoints from universality to pure noise.
>
> #### **7. About the strengths of our formulation.**
>
> Compared with prior methods, our proposed formulation achieves better results.
>
> The decoupling can also help us focus on the design of denosing network and condition encoding network respectively.
>
> In addition, our method can obtain great interpretability as shown in Figure 4, 6, 8 and 9. The class center can be also regarded as guidance without extra classifier or domain knowledge(Section 6.6).
>
> The intuition can also encourage faster sampling by choosing a earlier starting, which can be combined with the technique of strided sampling(Section 6.5 and Appendix D).
>
> #### **8. More related works.**
> The baseline "grad. DDPM" share the same formulation with [2*].
> And thanks for your advice. We will add those missing related works [1*, 2* ,3*] in the final revision.
>
> [1*] Improved Denoising Diffusion Probabilistic Models
>
> [2*] Diffusion Models Beat GANs on Image Synthesis
>
> [3*] ILVR: Conditioning Method for Denoising Diffusion Probabilistic Models
>
> #### **9. More experimental details.**
> The batch size is set to 128 for Cifar-10, 64 for LFW, and 24 for high-resolution Places2, LSUN, CUB-bird and Celea-hq.
> And thanks for your advice. We have already added  those missing details in the rebuttal revision.

---

> > ### Comment · Reviewer_HJtB · 2021-11-18
> > **Thanks for your reply**
> >
> > ES has been implicitly used in DDIM[4*], the author should cite it (The stride used DDIM doesn't start from T). Some response is still confusing. For example,
> > * In Q2, from my perspective, the first term $E[\frac{x_t - \sqrt{\alpha_t} x_0}{\sqrt{1-\alpha_t}}|x_t]$ ($x_0, x_t \sim q(x_0, x_t|c)=q(x_0|c)q(x_t|x_0,c)$) is a function of $x_t$ and $c$, instead of $x_t$ and $x_0$ claimed by the author. Since it can be explicitly written as $E_{q(x_0|x_t,c)}[\frac{x_t - \sqrt{\alpha_t} x_0}{\sqrt{1-\alpha_t}}]$, which is a function of $x_t$ and $c$.
> > * In Q3, the author doesn't answer whether $u=u_\phi(x_0)$ is a typo.
> > * In Q7, the author only talks about the proposed method instead of making a comparison.
> >
> >
> > [4*] Denoising Diffusion Implicit Models

---

> > > ### Author Response · Authors · 2021-11-18
> > > **Responses to Reviewer HJtB**
> > >
> > > Thanks for your comments and the related works will be added in the following revision.
> > >
> > > #### **Q2. About the equation.**
> > > If $x_t$ and $x_0$ are given, the first term can be computed directly. And we model the first term taking only the $x_t$ as input, which forces the denoising network to predict a reasonable $x_0$(e.g. a bird-like shape to generate a bird), and the class center $\mu$ in the second term is for guidance. More importantly, to predict $x_{t-1}$, the input $x_t$ has been modulated by $\mu$ and $x_{t+1}$ at $t+1$ . Actually, we have implicitly introduced the condition and the class center into the denoising network during the step-by-step reverse process.
> > >
> > > And definitely you are right. This could be another perspective by explicitly introducing the class center $\mu$ for the denoising network.
> > >
> > > #### **Q3. About the typo.**
> > > Thanks for your comments. We have realized that $\mu=\mu_\phi({x_0})$ is a typo.
> > > It should be corrected to $\mu=\mu_\phi({c})$, and we have already fixed this error in the rebuttal revision.
> > >
> > > #### **Q7. About prior methods.**
> > > Thanks for your comments. Prior methods can be divided into two categories.
> > >
> > > And the first is naive. The forward process remains unchanged and the condition is directly injected into the denoising network for prediction during reverse process. (e.g. concat). Compared with those methods, our method achieve better results and great interpretability.
> > >
> > > The second is given by [1]. With bayes rule, Song et al. implement controllable generation by introducing gradient guidance, which is given by extra time-dependent classifier or domain knowledge. Computationally, an advantage of the proposed approach over the guidance strategy is that no backpropagation through a classifier or other estimation is needed. More analysis is given in Section 6.6.
> > >
> > > [1] Score-based generative modeling through stochastic differential equations

---

> > > > ### Comment · Reviewer_HJtB · 2021-11-19
> > > > **Thanks for your reply**
> > > >
> > > > Can you specify more how to calculate the first term given $x_t$ and $x_0$? Do you implicitly assume that $c$ is a function of $x_0$?

---

> > > > > ### Author Response · Authors · 2021-11-19
> > > > > **Responses to Reviewer HJtB**
> > > > >
> > > > > Thanks for you comments. In our formulation, with $x_t$, the loss function is to minimize the MSE loss of noise prediction where the ground-truth noise is given by:
> > > > >
> > > > > $
> > > > > \epsilon_t = \frac{x_t - \sqrt{{\bar \alpha}_t}x_0}{\sqrt{1 - {\bar \alpha}_t}} - \frac{n_t}{\sqrt{1 - {\bar \alpha}_t}}
> > > > > $
> > > > >
> > > > > We can give a formulation by relocation, given by:
> > > > >
> > > > > $
> > > > > o_t = \epsilon_t + \frac{n_t}{\sqrt{1 - {\bar \alpha}_t}}
> > > > > = \frac{x_t - \sqrt{{\bar \alpha}_t}x_0}{\sqrt{1 - {\bar \alpha}_t}}
> > > > > $
> > > > >
> > > > > The right term is only related to $x_t$ and $x_0$. With $x_t$ and $x_0$, we can get the exact value of $o_t$.
> > > > > However, we only have $x_t$. The denoising network is exactly to learn a function to generate a reasonable $o_t$/$x_0$ based on $x_t$. That is also the key point of generative models and the subtlety of our formulation.
> > > > > With $o_t$ and $n_t$, we can compute the predicted noise and further sample the next $x_{t-1}$.
> > > > > And the class center is computed based on the given condition $c$.

---

> > > > > > ### Comment · Reviewer_HJtB · 2021-11-19
> > > > > > **Thanks for your reply**
> > > > > >
> > > > > > As claimed by the author, the loss function is to minimize the MSE loss of noise prediction.
> > > > > >
> > > > > > What I want to point out is that, the MSE loss has a well property: the optimal prediction function is the conditional expectation. Formally,
> > > > > >
> > > > > > Lemma 1*: $E[Y|X] \in \arg\min_{g(X)} E[ ||Y - g(X)||^2 ].$
> > > > > >
> > > > > > Please see https://stats.stackexchange.com/questions/71863/problem-with-proof-of-conditional-expectation-as-best-predictor for detailed derivation.
> > > > > >
> > > > > > If I don't misunderstand, the loss in Eq.(11) can be written as
> > > > > >
> > > > > > $$ E_t E_{c, x_0,x_t \sim q(c, x_0, x_t)} || \epsilon_t - \epsilon_\theta(x_t, u_\phi(c), t) ||^2 $$
> > > > > >
> > > > > > Here $\epsilon_t = \frac{x_t - \sqrt{\bar\alpha_t} x_0 }{\sqrt{1-\bar\alpha_t}} - \frac{n_t}{\sqrt{1 - \bar\alpha_t}}$, which can be calculated given $x_0$ and $x_t$ as claimed by the author.
> > > > > >
> > > > > > By applying Lemma1* to the loss in Eq.(11), we know the optimal noise predictor is also the conditional distribution
> > > > > >
> > > > > > $$ E[\epsilon_t|x_t, c] = E[ \frac{x_t - \sqrt{\bar\alpha_t} x_0 }{\sqrt{1-\bar\alpha_t}} - \frac{n_t}{\sqrt{1 - \bar\alpha_t}} |x_t, c] = E[\frac{x_t - \sqrt{\bar\alpha_t} x_0 }{\sqrt{1-\bar\alpha_t}} | x_t, c] - \frac{n_t}{\sqrt{1 - \bar\alpha_t}}$$
> > > > > >
> > > > > > The optimal noise predictor is decomposed into two terms. The first one is an expectation conditioned on $x_t$ and $c$.
> > > > > >
> > > > > > Thereby, if we use the decouple $\epsilon_\theta(x_t, u_\phi(c), t) = \epsilon_\theta(x_t, t) - \frac{n_t}{\sqrt{1 - \bar\alpha_t}}$ proposed in the paper, it seems that $\epsilon_\theta(x_t, u_\phi(c), t)$ can't reach the optimal predictor, since $\epsilon_\theta(x_t, t)$ is unrelated to $c$.
> > > > > >
> > > > > > Thereby, in Q2, I'm concerned whether the decoupling is reasonable.

---

> > > > > > > ### Author Response · Authors · 2021-11-19
> > > > > > > **Responses to Reviewer HJtB**
> > > > > > >
> > > > > > > Thanks for your comments and valuable thoughts.
> > > > > > >
> > > > > > > - Mentioned that during forward process, $x_t$ is actually related to $c$ and is a mixed representation (add) from $x_0$ and class center $u$ and during reverse process, $x_t$ comes from $\epsilon_\theta(x_{t+1}, t+1)$ and $\frac{n_{t+1}}{\sqrt{1-{\bar \alpha}_t}}$.
> > > > > > > In our formulation, the class center $u$ can directly make basic calculations with $x_t$ rather than using complex neural networks for modeling.
> > > > > > > The denoising network from the first term have already been class-aware during training procedure even if we do not input the class center explicitly. With the same initialized noise $x_T$ and different centers, we can get totally different reverse trajectories, making $x_t$ class-aware, as shown in Figure 5(Right).
> > > > > > > The network $\epsilon_\theta$ can even infer the corresponding class itself (e.g. a bird-like shape -> a bird).
> > > > > > > The evaluation results can also prove the effectiveness.
> > > > > > >
> > > > > > > - And if we explicitly input the class center for the first term, we don't really need the second term since we can directly model the whole equation $\epsilon_t = \frac{x_t - \sqrt{{\bar \alpha}_t}x_0 - n_t}{\sqrt{1 - {\bar \alpha}_t}}$. That makes no difference to the methods of first category and we can't get many interesting results, either.
> > > > > > >
> > > > > > > - From another perspective, the first term could be condition-free and the decoupled second term is considered as a reasonable guidance and show a similar trend with the gradient of classifier during the reverse process compared with score-based method[1](Eq. 14) using bayes rule (Section 6.6).
> > > > > > >
> > > > > > > [1] Score-based generative modeling through stochastic differential equations.
> > > > > > >
> > > > > > > -----
> > > > > > >
> > > > > > > A formal derivation about the reason why the denoising network $\epsilon_\theta$ have already been class-aware is given as follows:
> > > > > > >
> > > > > > > In our formulation, the second term is actually a guidance and we can rewrite the transition equation:
> > > > > > >
> > > > > > > $z_t = \sqrt{{\bar \alpha}_t}x_0 + \sqrt{1-{\bar \alpha}_t}\epsilon_t$
> > > > > > >
> > > > > > > $x_t = z_t + n_t$
> > > > > > >
> > > > > > > where $n_t$ is the condition in the paper, and once $c$ and $t$ are given, $n_t$ is fixed.
> > > > > > >
> > > > > > > Therefore, the target loss can be rewrite based on your theory as follows:
> > > > > > >
> > > > > > >  $E_t E_{c, x_0,z_t \sim q(c, x_0, z_t)} || \epsilon_t - \epsilon_\theta(z_t + n_t, u_\phi(c), t) ||^2 =
> > > > > > > E_t E_{c, x_0,z_t \sim q(c, x_0, z_t)} || \epsilon_t - \epsilon_\theta(x_t, u_\phi(c), t) ||^2 $
> > > > > > >
> > > > > > > By applying Lemma1*:
> > > > > > >
> > > > > > > $ E[\epsilon_t|z_t, c]
> > > > > > > = E[\frac{x_t - \sqrt{\bar\alpha_t} x_0 }{\sqrt{1-\bar\alpha_t}} | z_t, c] - \frac{n_t}{\sqrt{1 - \bar\alpha_t}}
> > > > > > > = E[\frac{x_t - \sqrt{\bar\alpha_t} x_0 }{\sqrt{1-\bar\alpha_t}} | x_t = z_t + n_t ] - \frac{n_t}{\sqrt{1 - \bar\alpha_t}}$
> > > > > > >
> > > > > > > Therefore, the denoising network $\epsilon_\theta$ have already been class-aware.

---

> > > > > > > > ### Comment · Reviewer_HJtB · 2021-11-20
> > > > > > > > **Thanks for the reply**
> > > > > > > >
> > > > > > > > I think the last equation in the derivation does not necessarily hold, but I'm generally satisfied with the intuitive interpretation. I have updated my score.

---

### Official Review · Reviewer_FjT6 · 2021-11-02

**Correctness:** 3
**Technical Novelty And Significance:** 2
**Empirical Novelty And Significance:** 2
**Recommendation:** 6
**Confidence:** 3

**Main Review:**

Strength -
1) The idea of doing controllable generation from diffusion models across different tasks by efficiently conditioning the model is interesting.
2) The paper shows strong conditional image generation results. Qualitative results for the image inpainting tasks also look good.
3) Extensive experiments across different tasks and datasets.

Weakness -
1) I am not convinced by the motivation of the paper. As the noise strength added in the initial stages of forward process is small and it grows over time, it is natural to expect the intra-class to inter-class ratio to remain stable in the initial stages of the forward process.
2) Adding on to the above point, the motivation around modeling class centers is not clear. The experiments do not reveal the intra-class variance to decrease during the forward process, but rather they remain stable. Why should then the forward process pull the data points towards the center?
3) The results of faster diffusion denoising do not convince me. By starting from earlier starting points around the center, I expected the FID for instance when using 700 steps to not drop from 3 to 7. As a baseline, what's the performance difference when we naively use 700 steps for the DDPM model instead of using 1000 steps?
4) I found the experiments around image inpainting hard to understand. Can the authors make it clearer as to how is the training done for it? Do you randomly mask out regions of the image during training at each iteration? How mc-uch portion of the image is masked out at max?
5) I couldn't find a comparison of the text to image generation against other works. Can the authors include the FID numbers for other related works?

**Summary Of The Paper:**

The paper observes the ratio of intra-class variance to the inter-class variance during different time steps of the forward process of diffusion models and notes that the ratio initially remains stable for the early steps of the process and then greatly increases when the data distribution finally gets converted to noise distribution. Taking inspiration from this class clustering process, the paper proposes to explicitly model the class center in the forward and the reverse process, whereby during the forward process the samples are pulled towards the center, before ultimately becoming noise. A separate conditioning network is used to model the centre. By using class as conditions for the conditioning network, the paper is able to perform class condition generation from a diffusion model and shows good image generation performance in terms of FID and Inceptions scores. The paper also shows good performance for other controllable image generation tasks such as image inpainting, attribute conditioned generation, and text to image generation by efficiently conditioning the diffusion model.

**Summary Of The Review:**

The paper achieves good empirical performance but needs more polishing around the motivation and some experimental details. I will be happy to increase my rating if my queries are clarified.

---

> ### Author Response · Authors · 2021-11-11
> **Responses to Reviewer FjT6**
>
> We thank the reviewer for the valuable feedback on improving the quality of our work.
> We are delighted that the reviewer admits the novelty of our method and the strong results of our experiments.
> Below we address specific questions and comments:
> #### **1&2. About the motivation.**
> Instead of gradually rising, the ratio first maintain stable and then increases significantly. And the stable class speration and clustering in the first place encourages us to explicitly introduce class centers to describe the common attribute of datapoints from the same class, and the guided diffusion process pull the datapoints together by eliminating the specificity.
> Based on the formulation, the class centers is to show the common
> characteristics.
>
> In addition, besides the interpretability as shown in Figure 4, 6, 8 and 9. the class center can be also regarded as guidance without using extra classifier or domain knowledge for reverse process(Section 6.6).
>
>
> #### **3. About the results of faster diffusion denoising.**
> Given the Eq.(7), we can get a transfer relation as follows:
>
> $
> \mathbf{x}_t = \sqrt{\bar{\alpha}_t}\mathbf{x}_0 + \mathbf{n}_t + \sqrt{\beta_t}\boldsymbol{\epsilon}_t
> $
>
> As $t$ increases, $\sqrt{\bar{\alpha}_t}$ get closer to zero, leading to
> an approximate transfer relation as follows:
>
> $
> \mathbf{x}_t \approx \mathbf{n}_t + \sqrt{\beta_t}\boldsymbol{\epsilon}_t
> $
>
> Therefore, selecting a earlier starting timestep is an approximate way for faster sampling. If $t$ is smaller, $\sqrt{\bar{\alpha}_t}$ is still a factor that cannot be ignored, leading to the decrease in FID. Also, we formulate the ST-DDPM with strided sampling in Appendix D.
> The earlier starting can be combined with strided sampling to achieve further speedup without harming the sample quality(e.g. 90 steps in Table 4).
>
> #### **4. About the experiments around image inpainting.**
> For masks, we use the Irregular Mask Dataset collected by [1], which contains 55,116 irregular raw masks for training and 24,866 for testing.
>
> During training, for each image in the batch, we first randomly sample a mask from 55,116 training masks, then perform some random augmentations on the mask, finally we use it to mask the image and get our class center for training. So the training masks are different all the time. The mask is irregular and may be 100% hole due to augmentations.
>
> During testing, we use 12,000 test masks sampled and augmented from 24,866 raw testing masks. These 12,000 masks are categorized by hole size according to hole-to-image area ratios (0-20%, 20-40%, 40-60%). Then we evaluate on 12,000 random test images by randomly assigning these masks to images without replacement. We calculate all quantative results for these 12,000 inpainting results. All these settings are same with previous works and baselines. And
> the condition encoding network is a U-Net architecture for pixel-to-pixel prediction.
>
> And thanks for your advice. We have already added  those details in the rebuttal revision.
>
> [1] Image Inpainting for Irregular Holes Using Partial Convolutions
>
> #### **5. About the comparison against other works of text to image generation.**
> The evaluation results are given as follows:
>
> |  Method   | FID  | IS |
> |  ----  | ----  | ----  |
> | StackGAN [1]  | 51.89 | 3.70  |
> | StackGAN++ [2]  | 15.30  | 3.82  |
> | FusedGAN [3] | - | 3.92 |
> | AttnGAN [4]  | - | 4.36 |
> | MirrorGAN [5]| - | 4.56 |
> | AGAN-CL[6]  | -  | 4.97 |
> | ST-DDPM  | 18.18 | 4.52  |
>
> The proposed method achieves comparable results with the state-of-the-art methods. And thanks for your advice. We have already added those results in the rebuttal revision.
>
> [1] H. Zhang, T. Xu, H. Li, Stackgan: Text to photo-realistic im- age synthesis with stacked generative adversarial networks, 2016
>
> [2] H. Zhang, T. Xu, H. Li, S. Zhang, X. Wang, X. Huang, D. N. Metaxas, Stackgan++: Realistic image synthesis with stacked generative adversarial networks, 2017
>
> [3] N. Bodla, G. Hua, R. Chellappa, Semi-supervised fusedgan for conditional image generation, 2018
>
> [4] T. Xu, P. Zhang, Q. Huang, H. Zhang, Z. Gan, X. Huang, X. He, Attngan:Attngan: Fine-grained text to image generation with attentional generative adversarial networks, 2017
>
> [5] T. Qiao, J. Zhang, D. Xu, D. Tao, Mirrorgan: Learning text-to-image generation by redescription, 2019
>
> [6] M. Wang, C. Lang, L. Liang, G. Lyu, S. Feng, T. Wang, Attentive generative adversarial network to bridge multi-domain gap for image synthesis, 2020

---

> > ### Comment · Reviewer_FjT6 · 2021-11-20
> > **Thanks for your reply**
> >
> > I would like to thank the authors for their reply.
> > My concerns around the motivation mentioned in point 1 still remain the same.  As the noise strength added in the initial stages of forward process is small and it grows over time, it is natural to expect the intra-class to inter-class ratio to remain stable in the initial stages of the forward process.
> > I would continue with my rating for now.

---

> > > ### Author Response · Authors · 2021-11-20
> > > **Responses to Reviewer FjT6**
> > >
> > > Thanks for your comments and we would like to give more explanation.
> > >
> > > As you said, it is natural to expect the intra-class to inter-class ratio to remain stable in the initial stages of the forward process, which leads to stable class separation. The stable class separation is what we attempt to reveal.  The motivation is quite intuitive in fact. In both RGB and feature space, the visualization results indicate the clustering.  We then introduce class centers to describe the class separation and common characteristics. The proposed diffusion process is not required to follow the changes of ratio.
> > >
> > > Also, if reasonable, the diffusion process can be totally defined by people themselves. The original diffusion process comes from the Langevin Dynamics (natural noise). In our formulation, besides the natural noise, the class center is another slowly-injected force, leading to a variant guided diffusion process. The formulated diffusion is quite different to the original one without guiding and the intra-class ratio can get smaller in our formulation.  And the detailed derivation actually make it self-contained.

---

### Official Review · Reviewer_Xxqm · 2021-11-02

**Correctness:** 4
**Technical Novelty And Significance:** 3
**Empirical Novelty And Significance:** 2
**Recommendation:** 6
**Confidence:** 3

**Main Review:**

< Strength >
1. This paper provides a new variant of a score-based generative model, and it can be attractive for many machine learning researchers.

2. This paper suggests a new interesting finding, the changes of the ratio of intra-class and inter-class.

3. From a new finding, this paper proposes a new class center-based algorithm for conditional image generation. The proposed model, ST-DDPM, shows a significant performance improvement on the FID score and Inception Score.

< Weakness and Questions >
1. The quantitative results are reported for the CIFAR-10 dataset only. There are qualitative results for other datasets such as FFHQ and LSUN, but there are no quantitative results for that kind of dataset.

2. This paper only reports the FID and inception score. I am curious about the results of negative log-likelihood.

3. There are many new score-based model research [1,2,3], and it will be interesting if further discussion will be added to the paper.

[1] Tae, Jaesung, Hyeongju Kim, and Taesu Kim. "EdiTTS: Score-based Editing for Controllable Text-to-Speech." arXiv preprint arXiv:2110.02584 (2021).

[2] Kim, Dongjun, et al. "Score Matching Model for Unbounded Data Score." arXiv preprint arXiv:2106.05527 (2021).

[3]  Tashiro, Yusuke, et al. "CSDI: Conditional Score-based Diffusion Models for Probabilistic Time Series Imputation." arXiv preprint arXiv:2107.03502 (2021).

**Summary Of The Paper:**

Score-based generative models show great performance on image generation tasks.
Score-based generative models are composed of the two-phase, forward process, and reverse process.
On the forward process, the score-based model gradually corrupts the data with noise, and the reverse process restores the image by denoising.
This paper measures the ratio of intra-class and inter-class, and they find that the class separation decreases sharply and the data distribution is totally covered by the noise.
From a new finding, this paper proposes a class center on both forward and reverse processes for a controllable generation.

**Summary Of The Review:**

Please see the weakness and questions section.

---

> ### Author Response · Authors · 2021-11-11
> **Responses to Reviewer Xxqm**
>
> We thank the reviewer for the thoughtful feedback on improving the quality of our work. Below we address specific questions and comments:
> #### **1. About the quantitative results for other datasets.**
> We provide more qualitative and quantitative results in Appendix **C, D and E**, including the FID socre on the inpainting dataset, and FID socre with faster sampling and more examples on multiple datasets.
>
> #### **2. About the results of negative log-likelihood.**
> The results of negative log-likelihood are presented in **Table 2 of Appendix C on Page 18**.
>
> #### **3. More related works.**
> Tae et al.[1] propose an off-the-shelf speech editing methodology based on score-based model without extra training.
> Kim et al.[2] improve the diffusion models by analyzing the model at zero diffusion time and add an easily applicable modification to the score function.
> Tashiro et al.[3] propose a probabilistic imputation method for time series imputation conditioned on observed values.
> And thanks for your advice. We have already added  those related works in the rebuttal revision.
>
> [1] Tae, Jaesung, Hyeongju Kim, and Taesu Kim. "EdiTTS: Score-based Editing for Controllable Text-to-Speech." arXiv preprint arXiv:2110.02584 (2021).
>
> [2] Kim, Dongjun, et al. "Score Matching Model for Unbounded Data Score." arXiv preprint arXiv:2106.05527 (2021).
>
> [3] Tashiro, Yusuke, et al. "CSDI: Conditional Score-based Diffusion Models for Probabilistic Time Series Imputation." arXiv preprint arXiv:2107.03502 (2021).

---

> > ### Comment · Reviewer_Xxqm · 2021-11-18
> > **Thank you for your response.**
> >
> > Thank you for your response.
> > 1. For Table2, could you explain the reasons that the proposed methods do not show the improvements on NLL.
> > 2. I already checked the qualitative results for other datasets in Appendix.
> > However, there are no quantitative results, and the experiment on CIFAR-10 alone might not enough to verify the proposed models.

---

> > > ### Author Response · Authors · 2021-11-18
> > > **Responses to Reviewer Xxqm**
> > >
> > > #### **1. About the NLL metric.**
> > > Thanks for your comments. The NLL for DDPM is only an upper-bound instead of exact value while we do have lower upper-bound(3.69 vs 3.72).
> > >
> > > In addition, the negative log-likelihood reflects the confidence score of models to generate the specific sample $x_0$ from the standard normal prior $p(x_T)$. This metric could be a way to show the fitting degree with the real data distribution(analogous to training loss) rather than a reflection of the sample quality. Even with different reverse trajectory, the model can generate a "good" sample.
> > >
> > > Also, same as the original DDPM, our method do not optimize the likelihood directly. The NLL metric is to present the fact that the diffusion models can achieve better FID and IS scores than likelihood-based models even without direct maximum likelihood training.
> > >
> > > #### **2. About the quantitative results.**
> > > Thanks for your comments. Actually, in Appendix C, Table 3 presents the evaluation results on the Place2 dataset,  while Table 4 presents the evaluation results on CUB-bird dataset, Table 5 presents the results of faster sampling on the CIFAR-10 dataset.

---

> > > > ### Comment · Reviewer_Xxqm · 2021-11-18
> > > > **Response**
> > > >
> > > > Thank you for your response.
> > > > I don't think that the experimental dataset and the baseline models are enough.
> > > > However, the novelty point of this paper is clear, and I will increase my score.

---

### Official Review · Reviewer_49RM · 2021-11-04

**Correctness:** 3
**Technical Novelty And Significance:** 3
**Empirical Novelty And Significance:** 3
**Recommendation:** 6
**Confidence:** 3

**Main Review:**

**Strengths**
* The method is well-motivated through empirical observations that are clearly set forth in the paper.
* The paper is written in a way that is accessible to a wide audience, yet contains detailed derivations in the appendix.
* I expect the paper to be interesting to a wide audience due to the wide range of tasks it considers, and separately due to the proposed method being more (easily) interpretable than existing score-based generative model.

**Weaknesses**
* The baseline conditional generation method considered in the paper (cond. DDPM) appears to be worse than it's unconditional counterpart (DDPM) in terms of the FID / IS metrics (Table 1). One would expect the conditional generative model would outperform the unconditional one by a large margin. It seems that the baseline may not have been properly tuned / optimised, and absent a well-tuned baseline it's difficult to judge the performance of ST-DDPM.
* Faster sampling experiments are not particularly strong, and miss important baselines (e.g., "Learning to Efficiently Sample from Diffusion Probabilistic Models" by Watson et al.). The proposed technique still requires 70%-95% of the denoising steps of the full generation chain - far more than competing methods or hand-crafted beta schedules.

**Minor comments**
* Was not clear what "ST" in ST-DDPM stands for. Is that "shift"?
* How were the 2D projects in Figures 2 and 3 made? If t-SNE or a similar dimensionality reduction method, please include hyper-parameters in the Appendix.
* The word "Appendix" was not consistently capitalized (noticed this for "appendix B", but there may be more).
* What is a "fire embedding" (Section 6.1)?
* Left plot of Figure 10 is somewhat trivial: skipping x% of the training steps gives you a 1 / (1 - x / 100) speed up. E.g. 1.43 = 1 / 0.7, 1.12 ~= 1 / 0.9. This comes from the fact that all denoising steps have the same runtime complexity.

**Summary Of The Paper:**

The authors propose a modification of DDPM for conditional generation, in which they explicitly model the conditional mean ("cluster center") in the forward and reverse diffusion processes. This modification is motivated by the empirical observation that for the large portion of the diffusion samples form several distinct cluster. The resulting model, ST-DDPM, shows competitive results on conditional image generation and inpainting; and enjoys interpretability by means of cluster center visualisation.

**Summary Of The Review:**

The paper provides an interesting and well-motivated modification of DDPMs for conditional generation, but needs stronger baselines to help assess the performance of the proposed modification.

---

> ### Author Response · Authors · 2021-11-11
> **Responses to Reviewer 49RM**
>
> We thank the reviewer for the valuable feedback on improving the quality of our work.
> We are delighted that the reviewer admits the novelty of our method and the value of our works.
> Below we address specific questions and comments:
> #### **1. About the conditional baselines.**
> Thanks for your comments. In pratice, it seems difficult to get the FID results in original DDPM paper even if we have similar IS score. For fair comparison to conditional baselines in our paper, we use the same experimental settings and training resources. The proposed method gets better results than the baselines and the original paper. And as you said, besides the quantitative results, the idea of introducing class centers, elegant modification and qualitative results are expected to be interesting to a wide audience.
>
> #### **2. About the faster sampling experiments.**
> In fact, the earlier starting is a natural derivation of our formulation by
> leveraging the approximate transfer relation $\mathbf{x}_t \approx \mathbf{n}_t + \sqrt{\beta_t}\boldsymbol{\epsilon}_t$. We also extend our formulation and conduct derivation further by using both the technique of earlier starting and strided sampling in Appendix D and obtain much more stronger results(e.g. 90 steps, FID=4.45). The evaluation results could be better than some related works[1]. And thanks for your advice. We have already added the missing related works for comparison in the rebuttal revision.
>
> [1] Learning to Efficiently Sample from Diffusion Probabilistic Models.
>
> #### **3. About minor comments.**
>
> Thanks for you comments and we have already clarified the following statements
> added the missing details into the appendix in the rebuttal revision.
> - As you said, "ST" stands for "shift".
> - The 2D projects are given by t-SNE that is initialized by PCA reduction.
> - A fire embedding stands for an initialized vector.
> - Actually, the denoising steps do have the same runtime complexity due to the input having the same shape and the same denoising network.

---

### Author Response · Authors · 2021-11-12
**About the rebuttal revision**

We thank all the reviewers so much for the valuable comments on improving the quality of our work. We have updated
the paper in the rebuttal revision as follows:
- Added some missing related works.
- Fixed some typos and statements to make our paper clearer.
- Added some missing experimental details. (See Appendix B)
- Added some baselines of text-to-image generation for further comparison. (See Table 4 in Appendix C)
- Added some missing derivation to make our paper clearer. (See Appendix D)

---

### Decision · Program_Chairs · 2022-01-20

**Decision:**

Reject

**Comment:**

The paper presents an interesting approach for defining conditional diffusion models. The core idea of this work is based on a new analysis of how class centers evolve in the forward diffusion process. On this positive side, this work builds on top of this analysis and introduces conditional diffusion processes that are guided towards class centers. This paper shows marginal improvements in small image datasets (MNIST and CIFAR-10) and auxiliary applications such as image inpainting and attribute-based image synthesis (demonstrated through only qualitative experiments). On the negative side, the proposed approach has the fundamentally limiting assumption that a class can be represented by a cluster center in the RGB space. Unfortunately, this assumption does not hold for practical datasets such as ImageNet where samples in each class have high diversity, and the class centers in the pixel space are not very distinct for different categories. The reviewers have rated this paper slightly above the borderline. They have acknowledged the novelty of the proposed guided diffusion. But they have criticized the submission for the lack of experiments on more common and challenging benchmarks. They also have criticized this work for not providing quantitative results on the auxiliary tasks. I agree with the reviewers that these experiments would shed light on whether the class center idea would hold for more challenging scenarios.

In the rebuttal, the authors provided additional quantitative results for the text-to-image generation task. However, these results show that the proposed method is outperformed by prior works. Most other auxiliary tasks including image inpainting and attribute-to-face generation are still demonstrated through qualitative experiments without detailed quantitative results.

In summary, given the limitation of the proposed approach, this submission currently lacks an in-depth analysis of the proposed work on challenging benchmarks and a detailed quantitative comparison to relevant baselines for the auxiliary tasks. Because of these concerns, we believe that the paper in its current form is not ready for publication at ICLR at this point.